# Convergence of aging- and rejuvenation-related epigenetic alterations on PRC2 targets

Oscar Camacho [ID] [1,9], Michael A Koldobskiy [ID] [1,2,9 ✉], Pradeep Reddy [ID] [3,8], Atharv Oak[1], Yuxiang Sun[4], Kenna Sherman[1,5], Juan Carlos Izpisua Belmonte [ID] [3,8 ✉] & Andrew P Feinberg [ID] [1,5,6,7 ✉]

## Abstract

**Rejuvenation of tissues in physiologically aging mice can be accomplished by long-term partial reprogramming via expression of reprogramming factors (Oct4, Sox2, Klf4, and c-Myc). To investigate the epigenetic determinants of partial reprogramming-mediated rejuvenation, we used whole-genome bisulfite sequencing to carry out unbiased comprehensive profiling of DNA methylation changes in skin from mice subjected to partial reprogramming, as well as young and untreated old controls. We found a striking convergence of age- and rejuvenation-related epigenetic alterations on targets of the Polycomb repressive complex 2 (PRC2), with increased DNA methylation level and entropy over these regions. Native ChIP demonstrated extensive loss of H3K27me3 in aged epidermis compared to young, partially overlapping regions with age- and rejuvenation-related DNA methylation changes. In addition, large H3K9me2-marked "LOCK" heterochromatin domains defined the boundaries for hypomethylated highly entropic regions during aging. These results are also supported by a likewise prominent enrichment of PRC2 targets in gene expression data, suggesting that PRC2 activity can modulate aging and mediate tissue rejuvenation.**

**Keywords** Aging; Rejuvenation; DNA Methylation; PRC2; LOCKs
**Subject Category** Chromatin, Transcription & Genomics

## Introduction

Partial reprogramming using cyclic expression of the transcription factors OCT4, SOX2, KLF4, and c-MYC (OSKM) has emerged as a powerful strategy to reverse age-associated phenotypes (Browder et al, 2022b; Lu et al, 2020; Ocampo et al, 2016; Rodriguez-Matellan et al, 2020; Wang et al, 2021). Recently, Browder et al established a long-term partial in vivo reprogramming protocol in normal physiologically aging mice, using mice carrying a single copy of an OSKM polycistronic cassette and a reverse tetracycline transactivator (rtTA) in a C57BL/6J genetic background (4F mice). Analysis of the skin of old treated mice revealed changes consistent with histologic and functional rejuvenation when compared to old untreated mice, including increased epidermal thickness, higher proliferative capacity, and decreased fibrosis after injury, reversal of age-related metabolic changes, and downregulation of genes involved in inflammation and epidermal differentiation (Browder et al, 2022b). This suggested that long-term partial reprogramming preserves a more plastic and less differentiated state in aged skin cells.

Given the dramatic reversal of aging-related signatures by long-term partial reprogramming, understanding epigenetic changes that underlie the rejuvenation process is essential. Previous application of a DNA methylation array-based aging clock, comprising fewer than 700 mostly non-functional CpG sites, indicated reversal of age-associated DNA methylation patterns following long-term partial reprogramming (Browder et al, 2022b; Horvath, 2013). However, besides the small number of CpG sites examined, epigenetic clocks are limited by the inherent challenge of distinguishing between biological and chronological aging (Bell et al, 2019).

Epigenetic alterations are now recognized not only as biomarkers of aging but also as plausible mechanistic mediators of age-associated tissue decline. One influential framework posits that mammalian aging reflects a progressive "loss of epigenetic information", the erosion of DNA methylation and chromatin programs that preserve cellular identity and function (Yang et al, 2023a). In line with this concept, a quantitative physical framework in which the epigenome is described as a potential energy landscape demonstrated that DNA methylation entropy increases with age (Jenkinson et al, 2017), reflecting a progressive flattening of epigenetic valleys that define cell type-specific identity.

Among chromatin regulators, the Polycomb repressive complex 2 (PRC2), which catalyzes histone H3 lysine 27 trimethylation (H3K27me3), has been increasingly associated with age-related epigenetic remodeling. Notably, PRC2-bound regions are hotspots for age-related DNA-methylation gain (Dozmorov, 2015; Moqri et al, 2024), suggesting that PRC2-regulated domains are particularly prone to age-related epigenetic drift. H3K27me3, the catalytic product of PRC2, is a hallmark of facultative heterochromatin and

[1]Center for Epigenetics, Johns Hopkins University School of Medicine, Baltimore, MD, USA. [2]Pediatric Oncology, Sidney Kimmel Comprehensive Cancer, Baltimore, MD, USA. [3]Salk Institute for Biological Studies, La Jolla, CA, USA. [4]Department of Nutrition, Texas A&M University, College Station, TX, USA. [5]Department of Genetic Medicine, Johns Hopkins University School of Medicine, Baltimore, MD, USA. [6]Departments of Medicine, Biomedical Engineering, and Public Health, Johns Hopkins University, Baltimore, MD, USA. [7]Department of Biomedical Engineering, Tel Aviv University, Tel Aviv, Israel. [8]Present address: Altos Labs, San Diego, CA, USA. [9]These authors contributed equally: Oscar Camacho, Michael A Koldobskiy. ✉E-mail: mak@jhmi.edu; jcbelmonte@altoslabs.com; afeinberg@jhu.edu

transcriptional repression, and its abundance and genomic distribution show context and tissue-dependent remodeling with age (Booth and Brunet, 2016).

In parallel, global loss of DNA methylation, particularly in CpG-poor and heterochromatic regions, is a well-documented feature of aging across multiple tissues (Jenkinson et al, 2017; Seale et al, 2022). In human epidermis, age- and sun exposure-related DNA hypomethylated domains were identified in extended genomic regions that partially overlap with H3K9me2-marked compartments known as LOCKs (Large Organized Chromatin K9-modifications) (Vandiver et al, 2015). LOCKs constitute repressive, lamina-associated chromatin domains that contribute to nuclear architecture and genome stability (Wen et al, 2009). The overlap between age-related DNA hypomethylated blocks and LOCKs highlights a potential link between DNA methylation drift and large-scale chromatin organization during aging.

Here we comprehensively characterized the genome-wide DNA methylation landscape in a model of controlled tissue rejuvenation induced by long-term partial reprogramming (Browder et al, 2022b). We performed whole-genome bisulfite sequencing (WBGS) comparing young, old, and OSKM-treated old skin, and used epigenetic landscape analysis, including measures of entropy as described (Jenkinson et al, 2017). Given prior work on chromatin remodeling and aging (De Lima Camillo et al, 2025; Yang et al, 2023b), we also performed native chromatin immunoprecipitation-sequencing (ChIP-Seq) on young and old purified epidermis, and compared the profiles of two heterochromatic marks, H3K27me3 and H3K9me2, to the age- and rejuvenation-related DNA methylation landscape analyses above.

# Results

## Global DNA methylation level and entropy shifts distinguish young, old, and rejuvenated skin

To investigate the DNA methylation landscape during aging and OSKM-mediated rejuvenation, we performed WGBS, which quantitatively profiles DNA methylation at single-base resolution across ~15 million CpG dinucleotides, providing over four orders of magnitude greater coverage than array-based epigenetic clocks. We carried out WGBS on whole skin (including dermis and epidermis) from five 4F mice treated with long-term partial reprogramming from 15 months of age until 22 months, four old untreated 4F mice (22 months), and three young 4F mice (3 months old) (Dataset EV1). WGBS data were analyzed using informME, a powerful tool for quantifying epigenetic variability by computing DNA methylation potential energy landscapes across the genome, capturing both mean methylation level (MML) and methylation variability as encapsulated by normalized methylation entropy (NME), a version of Shannon entropy that quantifies the disorder of methylation within an analysis region. This method permits identification of genomic regions of significant DNA methylation discordance using the Jensen–Shannon distance (JSD) of information theory, based on differences in probability distributions of methylation rather than conventional differential methylation analysis that is based solely on mean methylation differences (Jenkinson et al, 2018; Jenkinson et al, 2019; Jenkinson et al, 2017; Koldobskiy et al, 2021).

Principal component analysis (PCA) revealed a substantial separation between the different groups, capturing the age and rejuvenation-related differences in PC1 and the partial reprogramming treatment-specific effects in PC2 (Figs. 1A and EV1A). Genome-wide distributions of MML and NME showed a shift towards hypomethylation and a more stochastic and disordered epigenome in old untreated skin when compared to young skin (Fig. 1B,C). Conversely, old treated skin showed an increase in global methylation levels and a reduction of methylation entropy in comparison to old untreated samples, resembling young skin (Fig. 1B,D,E; Appendix Fig. S1). A similar trend was consistent when examining MML and NME distributions over selected genomic features, with the exception of CpG islands, which exhibit a trend toward increased MML with aging (Fig. EV1B).

Because whole skin comprises multiple cell types and DNA methylation differences may be driven by shifts in cell-type proportions, we performed a cell-type deconvolution analysis. In the absence of reference methylomes for mouse skin cell types, we used a reference-free approach implemented by the MeDeCom software (Lutsik et al, 2017), which decomposes the DNA methylation data into latent DNA methylation components (LMC) and their proportions in each sample (Fig. EV1C), with the goal of inferring cell-type proportions in the samples. The deconvolution results revealed that most samples were dominated by a single LMC, indicating relatively homogeneous DNA methylation patterns within samples and arguing against substantial cell-type admixture. Rather, aging or rejuvenation treatment was associated with a shift in the overall DNA methylation pattern, leading to a transition toward a different LMC. Importantly, since LMCs can be driven by biologically meaningful differences in DNA methylation as a result of aging or rejuvenation itself, a shift in the dominant LMC does not imply a cell-type composition shift.

## Epigenetic convergence on PRC2 target genes during skin aging and partial reprogramming

Given that aging and long-term partial reprogramming were associated with altered DNA methylation, we next sought to identify the genomic targets of epigenetic disruption in skin aging and rejuvenation. We ranked genes based on the potential of their DNA methylation states within their promoter and gene body regions (Jenkinson et al, 2019) to distinguish between young, old untreated, and old treated groups. The ranking was performed using the JSD magnitude, computed as the square root of the average of the sum of squares of all JSD values within analysis regions (Datasets EV2 and EV3). We identified the most epigenetically discordant genes and observed a very significant overlap between genes associated with age-related and rejuvenation-related changes (odds ratio (OR) = 16.95, $P = 2.2 \times 10^{-399}$). Specifically, 39.3% of the genes exhibiting discordance during aging were also present in the set of discordant genes linked to rejuvenation, identifying 661 genes that reversed age-related epigenetic changes through partial reprogramming. We then performed over-representation analysis (ORA) of the top 500 discordant genes identified in the aging and rejuvenation comparisons against the curated gene sets (C2 collection) from the MSigDB database (Liberzon et al, 2015). Remarkably, we found a high and consistent over-representation of PRC2 target genes, as

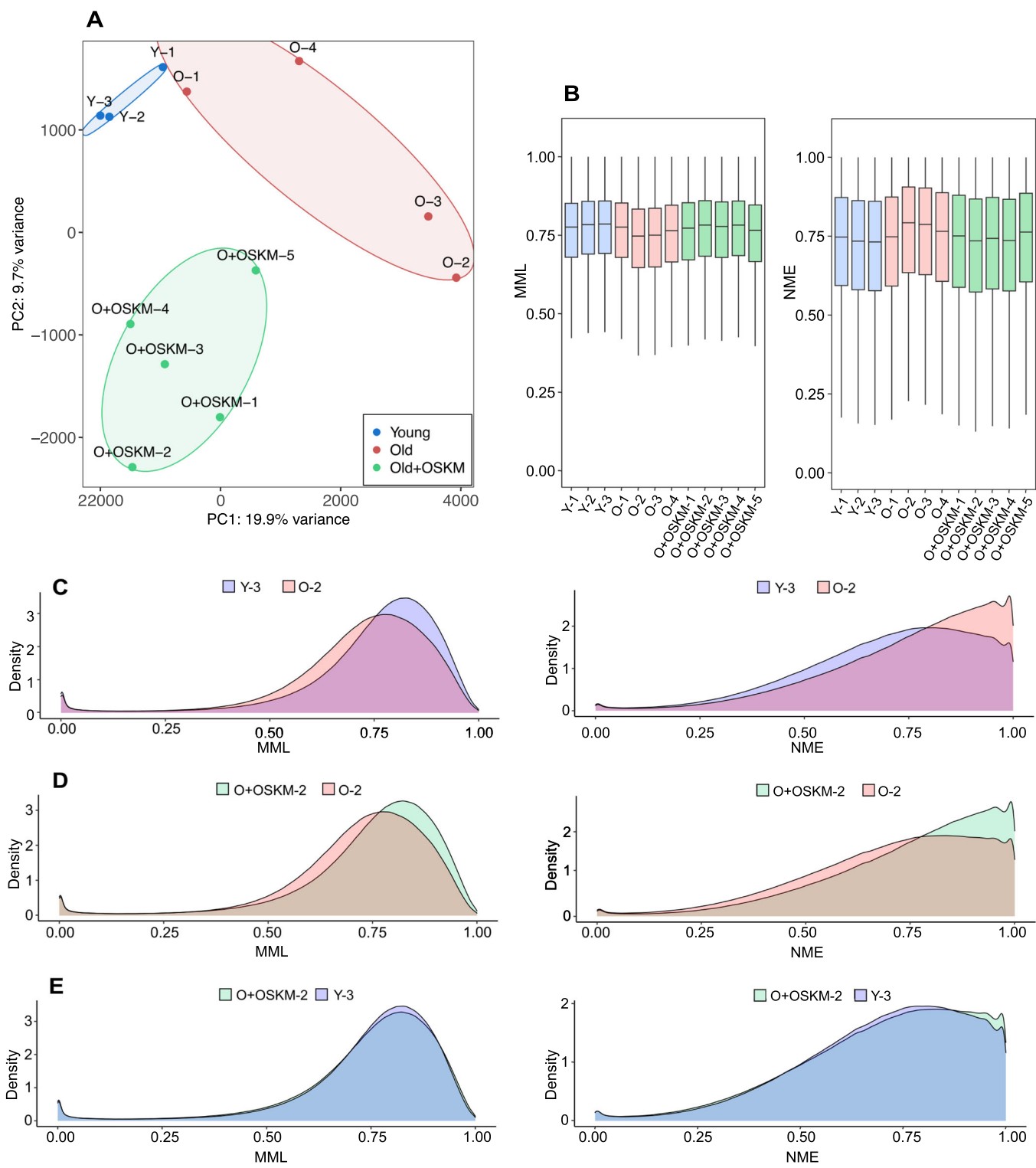

the top five over-represented gene sets in both comparisons consisted of genes annotated as bound by the PRC2 subunits EED and SUZ12 or marked by the H3K27me3 histone modification (Fig. 2A). These findings suggest a convergence of aging- and rejuvenation-related epigenetic changes specifically to genes that are targets of PRC2 activity. When assessing DNA methylation

changes in regions bound by EZH2, the catalytic subunit of PRC2, and H3K27me3-marked by employing available ChIP-Seq data from mouse embryonic stem cells, we observed a shift in MML towards hypermethylation and a substantial increase in NME in the old untreated samples when compared to young skin (Fig. 2C). Old treated skin exhibited a specific reversal of methylation changes in

◀

**Figure 1.  Age-associated genome-wide epigenetic shift is reversed with partial reprogramming-mediated rejuvenation.**

(A) PCA plot of genome-wide MML in WGBS samples from whole skin, including Young, Old untreated (Old) and Old treated (Old+OSKM) samples. (B) Box plots of genome-wide distributions of MML and NME. For each box plot, the central line represents the median, the box bounds correspond to the first (Q1) and third quartiles (Q3), and the whiskers extend to the most extreme data points within 1.5× the interquartile range (IQR) from the quartiles. To compare groups, complementary tests were performed: two-tailed and one-tailed Student's $t$ test (2T, 1T) on the average signal per sample, treating each biological replicate as one data point (Young, $n = 3$; Old, $n = 4$; Old+OSKM, $n = 5$); and a Wilcoxon signed-rank test (W) on tiled 150 bp genomic regions, where the signal was averaged across replicates within each group and compared tile-by-tile across the genome. All Wilcoxon signed-rank tests yielded $P < 2.2 \times 10^{-16}$, and pseudo-medians ($e$) are reported. MML comparisons: Old vs Young (2T: $P = 0.041$; W: $-0.018$); Old+OSKM vs Old (2T: $P = 0.053$, 1T: $P = 0.027$; W: $e = -0.017$). NME comparisons: Old vs Young (2T: $P = 0.042$; W: $e = 0.026$); Old +OSKM vs Old (2T: $P = 0.031$, 1T: $P = 0.016$; W: $e = 0.028$). (C–E) Density plots of genome-wide distributions of MML and NME in representative samples from each group.

these genomic regions. Interestingly, among the regions characterized in available chromHMM annotations, poised promoters, defined as the simultaneous presence of H3K27me3 and H3K4me3 histone marks, showed the greatest JSD and among the most elevated differences in NME in both young and old treated when compared to old untreated, indicating such regions exhibit pronounced epigenetic discordance, gain significant entropy with age and such epigenetic disorder can be reversed upon partial reprogramming treatment (Fig. EV2A). Furthermore, as opposed to the global age-related hypomethylation trend in other regions of the genome, poised promoters were the only domains exhibiting an age-related increase in DNA methylation, also reversed with partial reprogramming treatment.

To evaluate the relationship of these findings to gene expression changes during aging and partial reprogramming, we carried out differential gene expression analysis using published RNA-seq data from young, old untreated, and old treated skin samples (Data ref: Browder et al, 2022a; Browder et al, 2022b) (Datasets EV4 and EV5). Importantly, the PRC2-related gene sets that were top-ranked in the JSD-based analysis of differentially methylated genes were found likewise significantly over-represented among the differentially expressed genes (DEG) in both the aging and rejuvenation comparisons (Fig. 2B). Even though the overlap between DEG in aging and rejuvenation was strong (OR = 12.56, $P = 2.09 \times 10^{-109}$), the overlap between DEG and differentially methylated genes in aging and rejuvenation was modest (OR = 1.36, $P$ value $= 4.40 \times 10^{-4}$ and OR = 1.35, $P = 0.13$, respectively). Despite the small overlap, several genes found to be differentially methylated in both of our comparisons of aging and partial reprogramming were also differentially expressed in at least one of these processes. These included *Cdkn2a*, *Nr4a2*, *Egr3*, *Clcf1*, *Nrg1*, *Has1*, *SerpinB2*, *Clic5*, *Msrb3*, *Rbfox1*, multiple members of the Hox and Sox families, and other genes involved in metabolism, extracellular matrix remodeling, and inflammatory responses. Among the differentially methylated genes identified in both comparisons, but without significant changes in gene expression, we found *Rara*, a retinoic acid receptor known to reverse age-associated effects in skin (Mukherjee et al, 2006); *Errb4*, which has been positively correlated with keratinocyte proliferation and increased epidermal thickness (Hoesl et al, 2018); *Fgfr1*, a key regulator of keratinocyte migration and wound healing (Meyer et al, 2012); and *Cbx8*, a Polycomb Repressive Complex 1 component involved in the regulation of cellular senescence (Dietrich et al, 2007).

### Native ChIP-Seq reveals age-related loss of H3K27me3 in epidermis

Given the strong association between DNA methylation changes during aging and rejuvenation with PRC2 binding and H3K27me3-marked genes, we sought to directly examine H3K27me3 dynamics in aging in isolated epidermis. We purified the epidermal layer to minimize potential biases related to cell-type composition, and because prior work demonstrated that age-related epigenetic alterations in skin are most prominent in the epidermis (Vandiver et al, 2015). To achieve this, it was necessary to develop improved protocols for nuclear isolation and native ChIP-Seq, because of the low yield of immunoprecipitated material in conventional cross-linked ChIP, especially in old skin, for H3K27me3, and because of the increased signal-noise ratio of native ChIP for heterochromatin (Wen et al, 2009). Exogenous Drosophila chromatin was spiked in before immunoprecipitation for quantitative comparisons (ChIP-Rx) (Orlando et al, 2014). Differential peak analysis revealed a widespread loss of H3K27me3 across the genome in old specimens as compared to young (Figs. 3A and EV3A,B), with a significant reduction in with 44.7% of peaks at FDR < 0.05 and 12.3% remaining significant at the more stringent FDR < 0.01, a substantial fraction of which were located at gene promoters (Fig. 3B; Dataset EV6). Unless otherwise indicated, downstream analyses use the FDR < 0.01 threshold. Genes exhibiting differential H3K27me3 occupancy at promoters were over-represented in neuronal development pathways (Fig. EV3C), suggesting a loss of skin cell identity toward other neuroectodermal precursors than skin. Among the few genes that exhibited an increase in H3K27me3 within their promoter or gene body, we identified *Nr6a1*, a master regulator in vertebrate trunk development (Chang et al, 2022) and also implicated in wound healing in adult human skin (Wang et al, 2025); *Cecr2*, a histone acetyl-lysine reader involved in DNA damage response and γ-H2AX formation (Lee et al, 2012); and *Skida1*, which is suggested to modulate PRC2 activity (Liefke and Shi, 2015).

### Convergent H3K27me3 and DNA methylation alterations define a core PRC2-regulated gene set in aging and rejuvenation

Importantly, there was a significant overlap between genes displaying differential H3K27me3 peaks by native ChIP between young and old in isolated epidermis and differentially DNA methylated genes (significant JSD magnitude over the promoter and gene body) with aging (OR = 2.14, $P = 8.05 \times 10^{-19}$) and partial reprogramming (OR = 2.19, $P = 6.41 \times 10^{-18}$) in whole skin (epidermis and dermis combined). The association was stronger when the overlap was restricted to genes with promoter-localized changes in H3K27me3 and DNA methylation (aging: OR = 2.86, $P = 7.50 \times 10^{-16}$; partial reprogramming: OR = 3.25, $P = 7.00 \times 10^{-17}$). A further comparison was performed for Young vs Old and Old vs rejuvenated whole skin by stratifying promoters containing CpG islands according to their H3K27me3 dynamics (unmarked, retained, or marked with loss between the compared groups), and quantifying associated differences in CpG island–localized MML and NME. Among the promoters that exhibited an age-related loss of H3K27me3, there was a marked

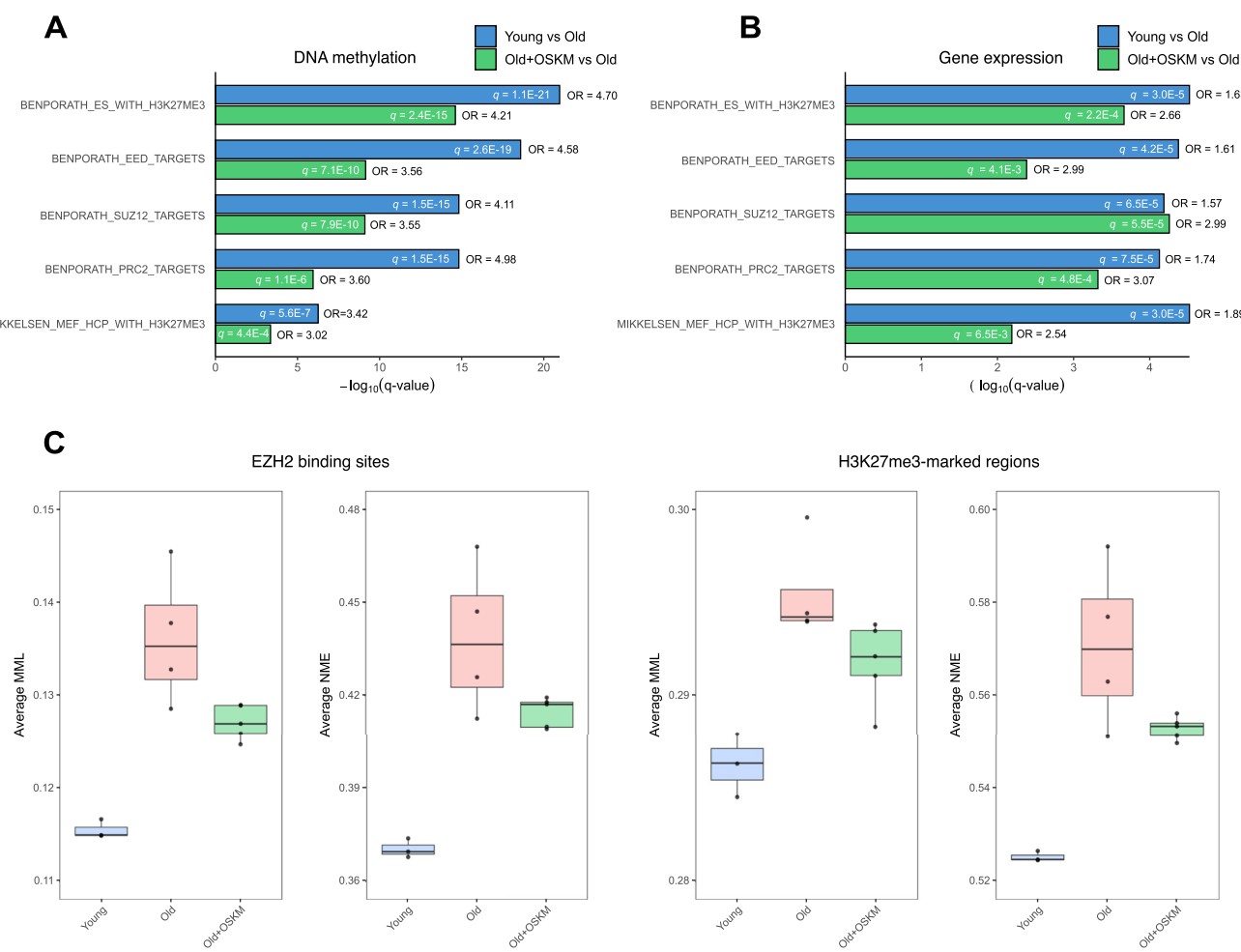

**Figure 2. DNA methylation discordance identifies PRC2 targets as major convergence elements in aging and partial reprogramming-mediated rejuvenation.**

(A) Top five over-represented MSigDB C2 gene sets identified using the top 500 differentially methylated genes ranked by Jensen–Shannon distance (JSD) in the Young vs Old (untreated) and Old+OSKM (treated) vs Old comparisons. (B) Over-representation of the (A) gene sets using differentially expressed genes in the Young vs Old and Old+OSKM vs Old comparisons. (C) Box plots of the average MML and NME per sample across 150 bp analysis units within EZH2 binding sites and H3K27me3-marked regions in mouse embryonic stem cells. For each box plot, the central line represents the median, the box bounds correspond to the first (Q1) and third quartiles (Q3), and the whiskers extend to the most extreme data points within 1.5× the interquartile range (IQR) from the quartiles. To compare groups, complementary tests were performed: two-tailed and one-tailed Student's *t* test (2T, 1T) on the average signal per sample, treating each biological replicate as one data point (Young, $n = 3$; Old, $n = 4$; Old+OSKM, $n = 5$); and a Wilcoxon signed-rank test (W) on the full set of tiled 150 bp genomic regions, where the signal was averaged across replicates within each group and compared tile-by-tile across the genome. All Wilcoxon signed-rank tests yielded $P < 2.2 \times 10^{-16}$ and pseudo-median ($e$) is reported. MML comparisons: EZH2 binding sites: Old vs Young (2T: $P = 0.005$; W: $e = 0.017$); Old+OSKM vs Old (2T: $P = 0.030$, 1T: $P = 0.015$; W: $e = 0.006$). H3K27me3-marked regions: Old vs Young (2T: $P = 0.004$; W: $e = 0.010$); Old+OSKM vs Old (2T: $P = 0.057$, 1T: $P = 0.029$; W: $e = 0.004$). NME comparisons: EZH2 binding sites: Old vs Young (2T: $P = 0.005$; W: 0.065); Old+OSKM vs Old (2T: $P = 0.067$, 1T: $P = 0.034$; W: $e = 0.022$). H3K27me3-marked regions: Old vs Young (2T: $P = 0.007$; W: $e = 0.043$); Old+OSKM vs Old (2T: $P = 0.059$, 1T: $P = 0.029$; W: $e = 0.017$).

increase in NME and a slight increase in MML in both aging and rejuvenation comparisons (Fig. EV3D).

We aimed to identify a core set of genes identified by our multimodal analysis as those with significant DNA methylation discordance (defined as significant JSD magnitude over promoter and gene body) during aging, significant DNA methylation discordance with partial reprogramming, and significant change in H3K27me3 distribution during aging. This produced an overlap set of 100 genes (Fig. 3C; Dataset EV7), representing a core set of PRC2-regulated genes that are epigenetically altered during aging and rejuvenation.

Prominent examples within this core set include *Sox11*, *Lin28b*, and *Galnt13* (Figs. 3D and EV3E). *Sox11* exhibits marked reduction in H3K27me3 over its promoter and gene body with aging, accompanied by methylation discordance, slightly increased MML, and markedly increased NME with aging, which are reversed with OSKM rejuvenation (Fig. 3D). *Sox11* is involved in epidermal development with key roles in wound repair (Miao et al, 2019) and skin fibrosis (Nanri et al, 2023), and has been implicated in aging phenotypes in other tissue types (Stevanovic et al, 2023). *Lin28b* shows the same pattern of convergent epigenetic changes as the previous example (Fig. EV3E) and has been identified as a central

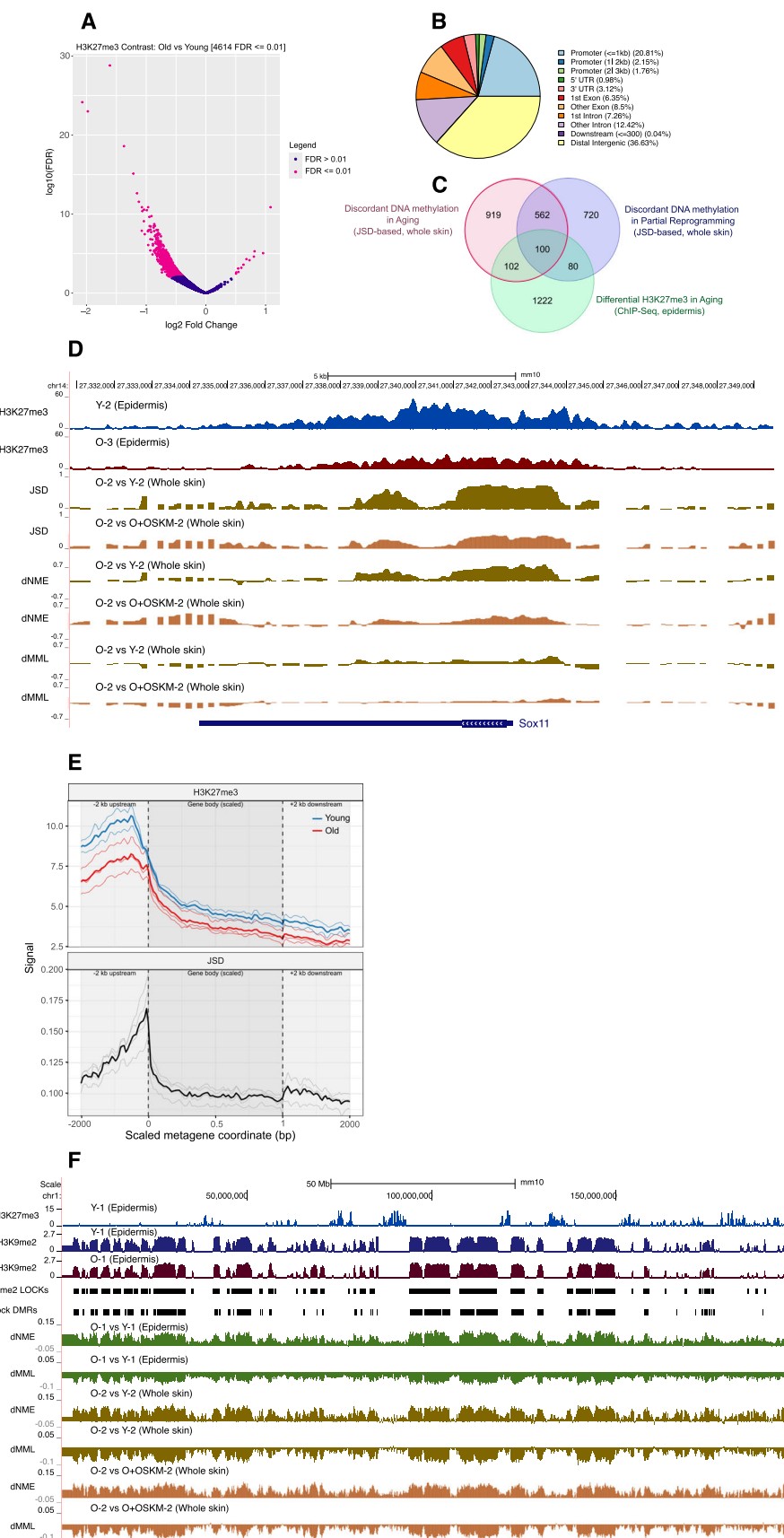

**Figure 3.  Native ChIP-Seq analysis demonstrates widespread H3K27me3 loss during skin aging.**

(A) Volcano plot of differential H3K27me3 peak analysis from ChIP-Seq comparing Old ($n = 3$) and Young ($n = 2$) samples. Differential H3K27me3 peaks (FDR < 0.01) are highlighted in pink. (B) Distribution of genomic regions harboring differential H3K27me3 peaks (FDR < 0.01). (C) Venn diagram showing the overlap of genes with differential DNA methylation (significant JSD magnitude over promoter and gene body) during aging and partial reprogramming in whole skin, and significant change in H3K27me3 occupancy during aging in purified epidermis. (D) Example of a gene (*Sox11*) exhibiting loss of H3K27me3 during aging colocalized to a region with increased DNA methylation discordance, as indicated by elevated JSD and dNME in old untreated (Old) samples compared to Young and Old treated (Old+OSKM) samples. (E) Metagene profiles of H3K27me3 occupancy from purified epidermis (top) and DNA methylation discordance measured as JSD from whole skin (bottom) during aging. Profiles were computed across the set of genes exhibiting significant aging-associated DNA methylation discordance (ranked by JSD magnitude over promoters and gene bodies) in whole skin. Gene bodies were scaled to a uniform length and flanked by ±2 kb of unscaled upstream and downstream sequence. Thin lines represent individual replicates for H3K27me3 ChIP, while for JSD they represent 4 comparisons (each old vs a representative young (Y-3) sample). Bold lines represent the average signal. This profile indicates coordinated promoter-proximal loss of H3K27me3 and elevated DNA methylation discordance with age. (F) Epigenomic landscape across chromosome 1 showing H3K27me3 and H3K9me2 occupancy, age-associated block DMRs defined in purified epidermis, and differential DNA methylation profiles (dMML and dNME) from purified epidermis and whole skin across Young, Old, and Old+OSKM samples. Age-related block DMRs overlap H3K9me2 LOCKs, defining regions of loss of DNA methylation and increase in entropy during aging that are partially reversed with partial reprogramming.

regulator of physiological aging, including heart (Jun-Hao et al, 2016), blood (Stolla et al, 2019) and testis (Sreejith et al, 2019), and has been shown to promote tissue repair when overexpressed in adult mice (Shyh-Chang et al, 2013). Members of this core set of genes exhibiting H3K27me3 alterations with aging and reversible changes to DNA methylation stochasticity during aging and rejuvenation warrant further study as mediators of aging-associated phenotypes. We also performed WGBS in old and young isolated tail epidermis (Fig. EV4A,B; Dataset EV8) and found a highly significant overlap (OR = 5.72, *P* value = 9.06 × $10^{-239}$ using all significant genes by JSD magnitude; OR = 14.98, *P* value = 3.43 × $10^{-200}$ using top 1000 ranked by JSD magnitude for both) between differentially methylated genes identified by JSD in the isolated epidermis and whole skin during aging, supporting the validity of the original data in whole dorsal skin.

To evaluate the relationship between DNA methylation alterations and H3K27me3 occupancy, we generated metaplots over promoter and gene body regions comparing H3K27me3 peaks and JSD between young and old from both whole skin and from purified epidermis comparisons (Figs. 3E and EV4C–G). Importantly, maximal H3K27me3 colocalizes with high JSD regions at gene promoters, where the pattern of aging-related H3K27me3 loss is most pronounced (Fig. 3E).

### DNA hypomethylated blocks overlap H3K9Me2 LOCKs

We previously identified large organized K9-modification (LOCK) domains marked by H3K9me2 (Wen et al, 2009). While age and sun exposure-related DNA hypomethylated blocks have been shown to overlap LOCKs, to our knowledge, there is no known association between differential DNA methylation and LOCKs in aging itself, and DNA methylation and H3K9me2 have not been measured directly in the same tissues in aging (Vandiver et al, 2015). To address this question, we used a contemporary methylation-block finding method ((Korthauer et al, 2019), see "Methods") on purified old and young epidermis, identifying 4707 block differentially methylated regions (DMRs) (*q* value < 0.05), 4705 of which were hypomethylated in old compared to young mice (Dataset EV9). We also performed native ChIP-Seq for H3K9Me2 and identified LOCKs with an average width of 1.3 Mb (Dataset EV10; Fig. EV5D,E). The LOCKs were largely unchanged during aging, with few (9/795; FDR < 0.05) differential regions. Remarkably, we found block DMRs and H3K9me2-marked LOCKs colocalized significantly (*P* value = 9.9 × $10^{-5}$, Z-score = 62.39): 4106/4707 (87.23%) of block DMRs overlapped a LOCK and 551/

795 (69.31%) of LOCKs overlapped at least one block DMR (Fig. 3F; also showing high degree of overlap with dNME), suggesting that LOCKs define regions where DNA methylation is lost in aging. There was no significant overlap between H3K27me3 peaks and H3K9me2 LOCKs (*P* = 1, *Z* = −97.77) or block DMRs (*P* = 1, *Z* = −64.22).

## Discussion

In summary, we report a novel analysis of the aging epigenome based on epigenetic discordance as captured by the JSD, identifying PRC2 domains as the most epigenetically disrupted regions during aging due to hypermethylation and significant gain of entropy. Most importantly, we show that long-term partial reprogramming treatment reverses such epigenetic disruption, decreasing both the mean methylation and entropy levels. We also found a striking convergence of age- and rejuvenation-related epigenetic alterations on PRC2 targets. These results are also supported by a likewise prominent enrichment of PRC2 targets in gene expression data, and loss of H3K27me3 during physiologic aging in mice. We have recently identified entropy as well as mean levels of DNA methylation to be associated with specific DNA binding motifs and regulatory DNA involved in developmental plasticity (Fang et al, 2023), indicating that such measures can unveil more comprehensive information than measures of mean differences alone.

Prior studies evaluating a limited number of CpG sites using custom mammalian methylation arrays in naked mole rats and dogs found an enrichment of H3K27me3 marks and PRC2 binding sites identified in stem cells, among genes proximal to hypermethylated age-related CpGs (Horvath et al, 2022a; Horvath et al, 2022b). A recent re-analysis of multiple published aging datasets supports an aging-related signature consisting of increased mean methylation of regions that exhibit low-methylation and PRC2 binding in embryonic stem cells (Moqri et al, 2024). However, those studies did not experimentally measure chromatin or DNA methylation entropy. Large blocks (>1 Mb) of H3K27me3 were identified in a study of aging murine liver (Yang et al, 2023b), which reversed on partial hepatectomy with regeneration, although that study did not examine DNA methylation in the tissues. We did not observe such blocks, but rather local loss of H3K27me3 at promoters, with loss of epidermal identity, and overlapping with observed differential DNA methylation entropy.

Consistent with the novelty of our study, there is relatively little overlap between the genes identified in those previous studies and the core gene sets identified here (Fig. EV5A–C). Our comprehensive, unbiased analysis of all CpG sites genome-wide, which also incorporates DNA methylation stochasticity, and relates these findings to H3K27me3 distribution on chromatin and gene expression, suggests a mechanistic role of altered PRC2 regulation in aging and rejuvenation.

From our novel native ChIP analysis, we also noticed a striking overlap of differentially methylated regions and H3K9me2 LOCK regions. While H3K9me2 LOCK regions were largely unchanged during aging, these regions showed striking overlap with blocks of age-related DNA hypomethylation, suggesting that LOCKs define regions of DNA methylation loss in aging. Interestingly, H3K27me3 and H3K9me2 regions were mutually exclusive, but both regions exhibited increased DNA methylation entropy with aging. This profound change of DNA methylation stochasticity at PRC2 sites and large-scale heterochromatin blocks in aging has parallels to cancer epigenomes, which are marked by large hypomethylated blocks and focal hypermethylated CpG islands (Feinberg et al, 2016).

We have previously proposed an information-theoretic model of aging wherein regions characterized by the highest entropic sensitivity represent deformable potential energy landscapes, corresponding to differentiation branch points (Jenkinson et al, 2017). After differentiation, regions associated with pluripotency/lineage plasticity have less need for maintenance of epigenetic fidelity, and may be more prone to age-associated epigenetic drift, leading to erosion of epigenetic information with aging. The identified enrichment of bivalent/poised regions among sites of greatest methylation change with aging may reflect the inappropriate opening of chromatin regions during aging. In all, our analysis of H3K27me3 alterations and DNA methylation entropy changes during aging and rejuvenation may suggest a mechanistic influence of PRC2 that warrants further investigation. Interestingly, repair of DNA damage in cultured fibroblasts was shown to increase H3K27ac and decrease H3K27me3 in genes related to disruption of cell identity (Yang et al, 2023a), and the authors speculate a relationship to epigenetic entropy. Another group made a prescient suggestion that the link between aging and cancer may involve hypermethylation and PRC2 (Jung and Pfeifer, 2015). Finally, the present study opens the door to efforts at PRC2 modulation to reverse aging, and identifies a core set of epigenetically convergent methylation and chromatin changes as specific targets to detect, modify, or reverse aging phenotypes.

# Methods

### Reagents and tools table

| Reagent/resource | Reference or source | Identifier or catalog number |
| --- | --- | --- |
| **Experimental models** | | |
| R26$^{rtTA}$;Col1a1$^{2lox-4F2A}$ (*M. musculus*) | The Jackson Laboratory | 011011 |
| C57BL6/J (*M. musculus*) | The Jackson Laboratory | 000664 |

| Reagent/resource | Reference or source | Identifier or catalog number |
| --- | --- | --- |
| **Recombinant DNA** | | |
| N/A | | |
| **Antibodies** | | |
| Rabbit anti-H3K27me3 | Cell Signaling Technology | 9733S |
| Mouse anti-H3K9me2 | Abcam | 1220 |
| Rabbit anti-H2Av (*Drosophila*) | Active Motif | 61686 |
| **Oligonucleotides and other sequence-based reagents** | | |
| N/A | | |
| **Chemicals, enzymes, and other reagents** | | |
| PBS | Thermo Fisher Scientific | 10010023 |
| Dispase type II | Thermo Fisher Scientific | 17105041 |
| Penicillin-Streptomycin | Thermo Fisher Scientific | 15140122 |
| DMEM high glucose | Thermo Fisher Scientific | 11965092 |
| MasterPure DNA Purification Kit | Biosearch Technologies | MC85200 |
| NEBNext Ultra II DNA Library Prep Kit for Illumina | New England Biolabs | E7645S |
| NEBNext Multiplex Oligos for Illumina | New England Biolabs | E7335S |
| Qubit dsDNA HS Assay Kit | Thermo Fisher Scientific | Q32851 |
| Ummethylated Lambda DNA | Promega | D1521 |
| AMPure XP beads | Beckman Colter | A63882 |
| EZ DNA Methylation-Gold Kit | Zymo Research | D5005 |
| Bioanalyzer 2100 High Sensitivity DNA Kit | Agilent | 5067-4626 |
| PhiX Control v3 Library | Illumina | FC-110-3001 |
| Kapa Hifi Uracil+ Kit | Roche | KK2801 |
| Kapa Library Quantification Kit | Roche | KK4824 |
| HEPES | Thermo Fisher Scientific | 15630080 |
| KCl | Millipore Sigma | 60142-100ML-F |
| MgCl$_2$ | Millipore Sigma | 63069 |
| Sucrose | Millipore Sigma | 8590-100 ML |
| Glycerol | Millipore Sigma | 49767-100 ML |
| DTT | Thermo Fisher Scientific | D1532 |
| PMSF | Cell Signaling Technology | 8553S |
| Tris-HCl | Millipore Sigma | 155568025 |
| Ca$_2$Cl | Millipore Sigma | C-34006 |
| Micrococcal nuclease | New England Biolabs | M0247S |
| EDTA | Thermo Fisher Scientific | R1021 |

| Reagent/resource | Reference or source | Identifier or catalog number |
|---|---|---|
| Triton X-100 | Millipore Sigma | 8590-100 ML |
| NaCl | Invitrogen | AM9760G |
| Sodium deoxycholate | Thermo Fisher Scientific | 89904 |
| SDS | Invitrogen | 15553027 |
| Spike-in Chromatin (*Drosophila*) | Active Motif | 53083 |
| Dynabeads Protein A | Thermo Fisher Scientific | 10001D |
| Dynabeads Protein G | Thermo Fisher Scientific | 10003D |
| LiCl | Millipore Sigma | L7026-100ML |
| $NaHCO_3$ | Millipore Sigma | S8761-100ML |
| Proteinase K | Biosearch Technologies | MPRK092 |
| RNase A | Biosearch Technologies | MRNA092 |
| ChIP DNA Clean & Concentrator | Zymo Research | D5205 |
| **Software** | | |
| TrimGalore v.0.6.10 | https://www.bioinformatics.babraham.ac.uk/projects/trim_galore | |
| FastQC v.0.11.2 | https://www.bioinformatics.babraham.ac.uk/projects/fastqc | |
| Bismark v.0.22.2 | https://www.bioinformatics.babraham.ac.uk/projects/bismark Krueger and Andrews, 2011 | |
| SAMtools v.1.19 | https://www.htslib.org Li et al, 2009 | |
| RnBeads v.3.21 | https://rnbeads.org Müller et al, 2019 | |
| MeDeCom v.1.0.2 | https://github.com/CompEpigen/MeDeCom Lutsik et al, 2017 | |
| informME v.0.3.3 | https://github.com/GarrettJenkinson/informME Jenkinson et al, 2017 | |
| dmrseq v.1.30 | https://github.com/kdkorthauer/dmrseq Korthauer et al, 2019 | |
| R v.4.4.2 | https://www.r-project.org/ | |
| Salmon v.1.9.0 | https://combine-lab.github.io/salmon Patro et al, 2017 | |
| tximport v.1.2.0 | https://www.bioconductor.org/packages/release/bioc/html/tximport.html | |
| DESeq2 v.1.30.1 | https://bioconductor.org/packages/release/bioc/html/DESeq2.html | |
| Bowtie2 v.2.4.2 | https://bowtie-bio.sourceforge.net Langmead and Salzberg, 2012 | |
| Picard v.3.4.0 | https://broadinstitute.github.io/picard | |
| BEDTools v.2.31 | https://bedtools.readthedocs.io | |

| Reagent/resource | Reference or source | Identifier or catalog number |
|---|---|---|
| Sambamba v.0.7.1 | https://lomereiter.github.io/sambamba | |
| deepTools v.3.5.0 | https://deeptools.readthedocs.io Ramírez et al, 2014 | |
| Sicer v.2.3 | https://zanglab.github.io/SICER2 Zang et al, 2009 | |
| EDD | https://github.com/CollasLab/edd Lund et al, 2014 | |
| DiffBind v.3.2.6 | https://bioconductor.org/packages/release/bioc/html/DiffBind | |
| fgsea v.1.36.0 | https://github.com/alserglab/fgsea | |
| regioneR v.1.42.0 | https://github.com/bernatgel/regioneR Gel et al, 2016 | |
| **Other** | | |
| Covaris microTUBEs | Covaris | 520216 |
| Covaris S220 Focused-ultrasonicator | Covaris | 500217 |
| 7900HT Real Time PCR System | ThermoFisher | 4329001 |
| GentleMACS C tubes | Milteny Biotec | 130-093-237 |
| GentleMACS Dissociator | Milteny Biotec | 130-093-235 |
| Qubit Fluorometer | ThermoFisher | Q32238 |
| 2100 Bioanalyzer Instrument | Agilent | G2939AAR |
| Magnetic separation rack | Invitrogen | 12321D |
| Thermocycler | Eppendorf | 6311000010 |
| Illumina HiSeq4000 System | Illumina | |
| Illumina NovaSeq X Plus System | Illumina | |

## Methods and protocols

### Animal use and care

All animal procedures were performed according to National Institutes of Health (NIH) guidelines and approved by the Committee on Animal Care at the Salk Institute. Mice carrying the OSKM polycistronic cassette and rtTA were obtained from The Jackson Laboratory (common name: R26[rtTA];Col1a1[2lox-4F2A], strain no. 011011). All the mice were in a C57BL/B6J background. Experimenters were not blinded to the treatment group. The protocols for cyclic induction of OSKM and collection of whole dorsal skin are described previously (Browder et al, 2022b). Briefly, the dorsal part of the mouse was shaved and sprayed with 70% ethanol, and a $3 \times 2$ cm rectangular piece of full-thickness skin was cut using forceps and surgical scissors. The whole dorsal skin piece was flash-frozen and stored in liquid nitrogen.

### Isolation of epidermis from tail skin

Tails from C57BL/B6J mice were collected, rinsed with 1× PBS (Sigma-Aldrich), and skin was separated from the bone using forceps after making a longitudinal incision. Isolated skin was incubated overnight with 4 mg/ml dispase II (Sigma-Aldrich) in DMEM (Thermo Fisher) supplemented with 5% penicillin/streptomycin (Thermo Fisher) in an end-over-end rotator at 4 °C. The following day, the tail skin was washed with 1× PBS and the epidermis was separated from the dermis using forceps, flash-frozen and stored in liquid nitrogen.

### DNA extraction, WGBS library preparation and sequencing

Genomic DNA was isolated from tissues using the MasterPure DNA Purification Kit (Biosearch Technologies). We confirmed the integrity of genomic DNA by gel electrophoresis. WGBS single-indexed libraries were generated using NEBNext Ultra II DNA Library Prep Kit for Illumina (New England Biolabs) according to the manufacturer's instructions with the following modifications: 500 ng input gDNA was quantified by Qubit dsDNA HS assay (Thermo Fisher) and spiked with 1% unmethylated Lambda DNA (Promega) to monitor bisulfite conversion efficiency. We fragmented input gDNA by Covaris S220 Focused-ultrasonicator to an average insert size of 350 base pairs (bp). Samples were sheared for 60 s using Covaris microTUBEs, with instrument settings of duty cycle 10%, intensity 5, and cycles per burst 200. Size selection was performed using AMPure XP beads and insert sizes of 300–400 base pairs were isolated. Samples were bisulfite converted after size selection using EZ DNA Methylation-Gold Kit (Zymo Research) following the manufacturer's instructions. After bisulfite conversion, we performed amplification using the Kapa Hifi Uracil+ (Roche) polymerase based on the following cycling conditions: 98 °C 45 s/8 cycles: 98 °C 15 s, 65 °C 30 s, 72 °C 30 s/72 °C 1 min. AMPure cleaned-up libraries were run on the 2100 Bioanalyzer High-Sensitivity DNA assay (Agilent) for quality control purposes. We quantified libraries by qPCR using the Kapa Library Quantification Kit for Illumina (Roche) and the 7900HT Real Time PCR System (Thermo Fisher). We sequenced WGBS libraries on an Illumina HiSeq4000 and NovaSeq X Plus instrument for whole skin and isolated epidermis samples, respectively, using 150 bp paired-end indexed reads and 5% of non-indexed PhiX library control (Illumina). Coverage and average CpG depth are included in Dataset EV1. The bisulfite conversion rate of unmethylated Lambda DNA was 99.5% on average.

### Native ChIP-Seq

Nuclei suspensions were prepared from 100 mg frozen epidermis by homogenizing the tissue in nuclei isolation buffer (10 mM HEPES, 10 mM KCl, 1.5 mM MgCl$_2$, 0.34 M sucrose, 10% glycerol, 1 mM DTT) using the GentleMACS Dissociator (Miltenyi Biotec) with program "4C_nuclei_1" at 4 °C. After centrifugation, the nuclei pellet was collected, and digestion of chromatin was performed by incubating 1 million nuclei with 400 U MNase in 200 µl of MNase buffer (50 mM Tris-HCl, 5 mM CaCl$_2$) for 10 min in a shaker (at 750 rpm) at 37 °C. The reaction was quenched with 10 mM EDTA, nuclei were lysed by adding 200 µL of nuclei lysis buffer (25 mM Tris-HCl, 150 mM NaCl, 1% Triton X-100, 1% sodium deoxycholate, 0.1% SDS), and the digested chromatin was solubilized by centrifuging at 18,000×g for 15 min at 4 °C. Supernatant was collected, and an aliquot of chromatin was used

to check digestion efficiency by BioAnalyzer DNA High Sensitivity (Agilent) assay and to quantify the amount of chromatin using Qubit HS assay (Thermo Fisher). In total, 2 µg of digested chromatin was used per immunoprecipitation reaction, and 10% of the amount was saved as input. Commercially available *Drosophila* chromatin (Active Motif) was used as spike-in for normalization purposes, following the recommendations of the manufacturer. Immunoprecipitation was performed overnight on a rotator at 4 °C using anti-H3K27me3 (Cell Signaling Technology) or anti-H3K9me2 (Abcam) antibody conjugated with 1:1 protein A:protein G Dynabeads (Thermo Fisher). The following day, immunoprecipitated complexes were washed 3X with low salt wash buffer (150 mM NaCl, 50 mM Tris-HCl, 0.1% Triton X-100, 1% SDS, 10 mM EDTA), 1× with high salt wash buffer (500 mM NaCl, 50 mM Tris-HCl, 0.1% Triton X-100, 1% SDS, 10 mM EDTA) and eluted by incubating 2× with elution buffer (100 mM NaHCO$_3$, 1% SDS) for 15 min at 65 °C. Eluted material was treated with RNase A and proteinase K (Thermo Fisher) overnight at 65 °C, and DNA was purified with DNA ChIP DNA Clean & Concentrator kit (Zymo Research). All buffers contained 1 mM PMSF. 5 ng of DNA was used to prepare libraries with NEBNext Ultra II library kit with unique dual index primers (New England Biolabs). The library quality and quantity were verified by BioAnalyzer DNA High Sensitivity assay. Libraries were pooled and paired-end sequenced on the NovaSeq X Plus platform (Illumina).

### Mapping and quality control of WGBS

FASTQ files were processed using Trim Galore v.0.6.10 to perform single-pass adapter- and quality-trimming of reads. FastQC v.0.11.2 was employed for quality control of reads. Reads were aligned to the mm10 genome (GENCODE release: M23) using Bismark v.0.22.2 (Krueger and Andrews, 2011). Separate M-bias plots for read 1 and read 2 were generated by running the Bismark methylation extractor using the "mbias_only" flag, and these plots were used to determine how many bases to remove from the 5′ end of reads. The amount of 5' trimming from read 1 and read 2 was 5 and 20 bp, respectively. BAM files were subsequently processed with SAMtools v.0.1.19 (Li et al, 2009) for sorting, merging, duplicate removal, and indexing.

### Cell type deconvolution from WGBS data

BED methyl files from whole skin WGBS samples were generated from deduplicated BAM files using the function "methylation_extractor" from Bismark v.0.22.2 (Krueger and Andrews, 2011) and used as input for downstream analyses. DNA methylation data was processed and filtered using RnBeads v.3.21 (Müller et al, 2019). All sites having read coverage lower than 5 in any of the samples, high and low coverage outliers, sites with missing values, annotated SNPs, and sites on the sex chromosomes were removed. In the absence of reference DNA methylation data for cell types in mouse skin, we employed MeDeCom v.1.0.2 (Lutsik et al, 2017), a reference-free computational framework that uses regularized non-negative matrix factorization to decompose complex DNA methylation data into latent methylation components (LMC) and their proportions in each sample. We selected the 5000 most variably methylated CpGs across the samples for this analysis. Since we did not have prior knowledge of the expected number of underlying cell types to select, we resorted to the cross-validation procedure of MeDeCom. We chose 7 LMCs as the value of *K* at

which the cross-validation error started to level out and selected $\lambda = 0.001$ (regularization parameter) as the point where the cross-validation error is still low, while the objective value and RMSE substantially change, following the guidelines described by the authors of the MeDeCom software (Lutsik et al, 2017).

### Computation of DNA methylation potential energy landscapes

We computed DNA methylation potential energy landscapes from WGBS data using informME v.0.3.3 (Jenkinson et al, 2017), a freely available information-theoretic pipeline for DNA methylation analysis based on the 1D Ising model of statistical physics. For these computations, the entire genome was partitioned into consecutive non-overlapping genomic windows of 3 kb each, and a potential energy landscape was estimated within each window from available WGBS reads using a maximum-likelihood approach (Jenkinson et al, 2018).

### Computation and analysis of mean methylation level and normalized methylation entropy

We further partitioned each 3-kilobase estimation window into 20 non-overlapping analysis regions of size 150 bp each. Within each 150 bp analysis region, we computed the probability distribution of the methylation level, the mean methylation level (MML), and the normalized methylation entropy (NME) directly from the associated potential energy landscape using informME (Jenkinson et al, 2018). The mean methylation level is the expected value of the methylation level $L$ within an analysis region, and is given by $E[L] = \sum_l l \times P_L(l)$, where $P_L(l)$, $l = 0, \frac{1}{N}, \frac{2}{N}, \ldots, 1$, is the associated probability distribution of $L$. The normalized methylation entropy is a normalized version of Shannon's entropy, given by $h = -[1/\log_2(N+1)] \sum_l P_L(l) \log_2 P_L(l)$, and was used to quantify the amount of methylation stochasticity observed within an analysis region. It ranges between 0 and 1, taking its maximum value when all methylation levels within an analysis region are equally likely (fully stochastic methylation), and achieving its minimum value only when a single methylation level is observed (perfectly ordered methylation). Differences in MML and NME between experimental conditions, assessed either genome-wide or across selected genomic regions, were evaluated using two complementary statistical analyses. First, we performed independent Student's $t$ tests using one value per sample, obtained by averaging the MML or NME values across all 150-bp analysis regions genome-wide (or across tiles within the selected genomic regions). These per-sample averages were then compared between conditions, treating each mouse as an independent biological replicate (Young: $n = 3$; Old untreated: $n = 4$; Old treated: $n = 5$). Comparisons between young and old untreated mice were evaluated with two-sided tests. For comparisons between old untreated and old treated mice, we report both two-sided and one-sided $P$ values, with the one-sided tests reflecting our a priori expectation that OSKM treatment would reverse age-associated changes. Second, to evaluate the consistency and directionality of methylation changes across the genome, we performed a paired Wilcoxon signed-rank test on 150-bp analysis regions. For each region, the average MML or NME within each condition was calculated by aggregating across biological replicates, and the resulting per-region values were compared in a paired design. We report the Wilcoxon pseudo-median ($e$), which provides a robust estimate of the central tendency of the paired differences in MML or NME across 150-bp analysis regions. Analysis of the differences in CpG island-specific MML and NME of promoters stratifying by status of the H3K27me3 mark in the promoter (Fig. EV3D), was done as follows. For each promoter (defined as a 4-kb window centered at the transcription start site) in each sample, we computed sample-level means by averaging MML and NME derived from 150 bp analysis regions that overlapped CpG islands. Promoters without CpG islands were not included. Then we computed condition-level means by averaging the sample-level means from each promoter, and calculated the difference of condition-level means for the relevant comparisons (Old–Young and Old– Old+OSKM). We classified promoters as having an increase or decrease in MML or NME in CpG islands if the difference between the condition-level means was greater or lower than 5%, respectively.

### Jensen–Shannon distance

Within a 150 bp analysis region, the JSD between the two probability distributions $P_L^{(t)}$ and $P_L^{(r)}$ of the methylation level in a test (i.e., "old") and a reference (i.e., "young") sample was calculated by $\sqrt{\frac{1}{2}[D_{KL}(P_L^{(t)}, \overline{P_L}) + D_{KL}(P_L^{(r)}, \overline{P_L})]}$, where $\overline{P_L} = \frac{P_L^{(t)} + P_L^{(r)}}{2}$ and $D_{KL}(P, R) = \sum_l P(l) \log_2 [\frac{P(l)}{R(l)}]$ is the Kullback–Leibler divergence between two probability distributions $P$ and $R$ (also known as the relative entropy), and was used to quantify dissimilarities between the two probability distributions $P_L^{(t)}$ and $P_L^{(r)}$. It ranges between 0 and 1, taking its minimum value only when the two probability distributions are identical, in which case no statistical discordance in methylation level is present, and its maximum value of 1 only when the supports of the two probability distributions do not intersect each other, in which case a maximum statistical discordance in methylation level is observed.

### Hypothesis testing and gene ranking based on JSD

As previously described (Jenkinson et al, 2019), in each test versus reference comparison, promoter and gene body regions were assigned a score given by the JSD magnitude, computed as $\sqrt{1/K \sum_{k=1}^{K} [\text{JSD}(k)]^2}$, where $\text{JSD}(k)$ is the JSD within the $k$th analysis region overlapping the genomic region of interest (promoter or gene body). One-sided hypothesis testing was performed by testing against the null hypothesis that a genomic feature exhibits a JSD magnitude that can be explained by normal technical, statistical, or biological variability. To do so, a null distribution was constructed for the JSD magnitude by comparing the corresponding reference samples (using the young samples in the "Old vs Young" test, and the old samples in the "Old+OSKM vs Old" test). To account for variability in the number of analysis regions overlapping each feature, generalized additive models were employed for location scale and shape (GAMLSS) with a logit-skewed Student's $t$ distribution. To evaluate genes in multiple test/reference comparisons, we used Fisher's summary test statistic to test the null hypothesis that epigenetic discordance observed within a gene's promoter and body in the test/reference comparisons is only associated with biological, statistical, or technical variability in the reference samples, against the alternative hypothesis that this discordance is due to other factors within at least one of the two

genomic features considered (promoter or gene body) in at least one of the test/reference comparisons, and followed a similar approach to evaluate genes using only their promoters or bodies. We then scored each gene by using the computed $P$ value for rejecting the null hypothesis and produced a ranked list of genes with increasing $P$ values, breaking possible ties by combining the $P$ value rankings obtained from every single test/reference comparison using the method of rank products. Finally, we evaluated the statistical significance of each ranking while controlling for the false-discovery rate at 0.05 using adjusted $P$ values computed by the Benjamini–Hochberg procedure.

### Differentially gene expression analysis

FASTQ files containing RNA-seq reads were mapped to the mm10 genome (GENCODE release: M23), and transcript-level quantification was performed using Salmon v.1.9.0. We used tximport v.1.2.0 to compute normalized gene-level counts from the transcript-level abundance estimates (scaling these using the average transcript length over samples and the library size). Differential gene expression was calculated using DESeq2 v.3.15, with multiple hypothesis correction of $P$ values performed using the Benjamini–Hochberg method. DESeq2's default independent filtering was applied to remove genes with extremely low counts prior to multiple testing. Log2 fold changes were shrunk using DESeq2's empirical Bayes method, and the shrunken values were used to apply the fold-change threshold, as recommended for stable fold change estimation (Love et al, 2014). For a gene to be called differentially expressed gene it required to have a Benjamini–Hochberg adjusted $P$ value < 0.05 with shrunken $\log_2$ fold change < -1.5 or >1.5.

### ChIP-Seq analysis

FASTQ files containing ChIP-Seq reads were trimmed to remove adapter sequences with TrimGalore v.0.6.1, and sequencing quality was assessed using FASTQC v.0.11.2. Reads were aligned to the mm10 (mouse; GENCODE release: M23) and dm6 (*Drosophila*: release 6) genome assemblies using bowtie2 v.2.4.2 (Langmead and Salzberg, 2012) with the end-to-end parameter. SAM output files were then filtered to retain uniquely mapped reads with a minimum mapping quality of 10 using SAMtools v.1.19 (Li et al, 2009), and PCR duplicates were removed using Picard v.3.4.0. Reads mapping to the Encyclopedia of DNA Elements (ENCODE) blacklisted regions were also removed from the analysis (Amemiya et al, 2019). bigWig files were generated using the bamCoverage function from deepTools v.3.5.0 (Ramírez et al, 2014), and input reads were subtracted from H3K27me3 and H3K9me2 signal with bigwigCompare. H3K27me3 and H3K9me2 peaks were called using Sicer2 v.2.3 (Zang et al, 2009) and EED (Lund et al, 2014), respectively, with the default parameters and using input as background. Differential peak analysis was performed using DiffBind v.3.2.6. Spike-in normalization was used in the analysis by dividing ChIP signal by the number of reads mapped to the *Drosophila* genome, as previously described (Egan et al, 2016). For visualization, bigWig files were scaled based on the spike-in normalization factors (Love et al, 2014; Patro et al, 2017; Soneson et al, 2016).

### Detection of block differentially methylated regions

Files with methylation data per CpG site were generated from sorted, merged, and deduplicated BAM files (described above) using bismark_methylation_extractor from Bismark v.0.22.2 (Krueger and Andrews, 2011). Data were subsequently imported into R as a bsseq object using the read.bismark function (Hansen et al, 2012). The bsseq object was filtered for CpGs with at least 2× coverage in at least four samples of both groups. Block DMRs were identified with dmrseq (Korthauer et al, 2019) using the following parameters: cutoff=0.05, block=TRUE, minInSpan=500, bpSpan=5e4, maxGapSmooth=1e6, maxGap=5e3, and adjusted $P$ value < 0.05 and at least 100 CpGs per block DMR.

### Genomic features and annotations

Files and tracks bear genomic coordinates for mm10. We used the R package "TxDb.Mmusculus.UCSC.mm10.knownGene" to define genes, exons, and introns. We obtained CpG island annotations from the UCSC Genome Browser. We defined the promoter region of a gene as the 4-kb window centered at its transcription start site and determined the gene body region to be the remainder of the gene. We used publicly available datasets containing coordinates for H3K27me3 (Data ref: ENCODE Project ENCSR000CFN, 2012) and EZH2 (Aljazi et al, 2020a; Data ref: Aljazi et al, 2020b) binding, and chromHMM annotations (Data ref: ENCODE Project ENCR684CDN, 2013) in mouse embryonic stem cells.

### Over-representation analysis

Over-representation analysis (ORA) was performed using the fora function from the fgsea R package (v3.22), which applies Fisher's exact tests to evaluate whether a given gene list is over-represented within curated gene sets (C2 collection) from the MSigDB database (Liberzon et al, 2015). For the ORA of differentially methylated genes, we used the top 500 genes ranked by JSD magnitude (and significant JSD-based epigenetic discordance; adjusted $P < 0.05$) and tested their over-representation across all the gene sets from the C2 collection. The background (universe) was defined as all genes considered in the ranking by JSD magnitude (Datasets EV2 and EV3), corresponding to the complete set of genes annotated in the TxDb.Mmusculus.UCSC.mm10.knownGene package. For the ORA of differentially expressed genes, genes with adjusted $P < 0.05$ and shrunken $\log_2$ fold change < $-1.5$ or >1.5 in the DESeq2 analysis were tested for over-representation among the top five gene sets from the C2 collection that ranked highest in the JSD-based ORA. The background was defined as all genes captured by RNA-seq and retained in the differential expression analysis with DESeq2 (Datasets EV4 and EV5).

### Overlap analysis of gene sets and genomic regions

Significance of overlap between gene lists was performed using Fisher's exact test. When comparing gene lists derived from different sequencing technologies (e.g., overlap between JSD-based differentially methylated genes and genes with differential H3K27me3 occupancy), the background was defined as the common set of genes assessed in both analyses (e.g., genes within sex chromosomes were excluded from overlaps involving JSD-based rankings because informME does not report values for those chromosomes). Significance of overlap between genomic regions was performed using the function permTest from the R package regioneR, which performs permutation test analysis. In total, 10,000 permutations were used. Z-score, defined as the distance between the expected value and the observed one, measured in standard deviations, was provided.

## Data availability

The datasets and computer code produced in this study are available in the following databases: WGBS data: Gene Expression Omnibus GSE231658. ChIP-Seq data: Gene Expression Omnibus GSE288574. Computer code produced in this study: Github (https://github.com/ocamach1/PRC2_aging_rejuvenation). Source computer code for informME: Github (https://github.com/GarrettJenkinson/informME).

The source data of this paper are collected in the following database record: biostudies:S-SCDT-10_1038-S44320-026-00195-9.

## Peer review information

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

## Acknowledgements

This work was supported by the Bloomberg Philanthropies (APF), and Johns Hopkins University funding of the XDBio Graduate Program (to OC). The funders had no role in study design, data collection and analysis, decision to publish, or preparation of the manuscript. We acknowledge the service provided by the High Throughput Sequencing Lab at Johns Hopkins School of Medicine.

## Author contributions

**Oscar Camacho**: Conceptualization; Data curation; Formal analysis; Investigation; Methodology; Writing—original draft; Writing—review and editing. **Michael A Koldobskiy**: Conceptualization; Data curation; Formal analysis; Supervision; Investigation; Methodology; Writing—original draft; Writing—review and editing. **Pradeep Reddy**: Resources; Investigation. **Atharv Oak**: Investigation; Methodology. **Yuxiang Sun**: Resources. **Kenna Sherman**: Investigation; Methodology; Writing—review and editing. **Juan Carlos Izpisua Belmonte**: Conceptualization; Resources; Supervision. **Andrew P Feinberg**: Conceptualization; Supervision; Funding acquisition; Methodology; Writing—original draft; Project administration; Writing—review and editing.

Source data underlying figure panels in this paper may have individual authorship assigned. Where available, figure panel/source data authorship is listed in the following database record: biostudies:S-SCDT-10_1038-S44320-026-00195-9.

## Disclosure and competing interests statement

PR and JCIB are currently employees of Altos Labs, but were not when materials were provided.

# Expanded View Figures

**Figure EV1.  Global DNA methylation patterns and sample composition in whole skin WGBS.**

(**A**) PCA plot of genome-wide MML in WGBS samples from whole skin, including Young, Old untreated (Old) and Old treated (Old+OSKM) samples. (**B**) Box plots of MML and NME distributions within selected genomic regions. For each box plot, the central line represents the median, the box bounds correspond to the first (Q1) and third quartiles (Q3), and the whiskers extend to the most extreme data points within 1.5× the interquartile range (IQR) from the quartiles. Student's *t* tests were performed using the average MML or NME value of each independent sample and comparing groups Young (*n* = 3), Old (*n* = 4) and Old+OSKM (*n* = 5). MML comparisons: Island: Old vs Young: *P* = 0.325; Old+OSKM vs Old: *P* = 0.074. Shore: Old vs Young: *P* = 0.129; Old+OSKM vs Old: *P* = 0.395. Shelf: Old vs Young: *P* = 0.037; Old+OSKM vs Old: *P* = 0.076. Open Sea: Old vs Young: *P* = 0.040; Old+OSKM vs Old: *P* = 0.047. Promoter: Old vs Young: *P* = 0.046; Old+OSKM vs Old: *P* = 0.146. Gene Body: Old vs Young: *P* = 0.037; Old+OSKM vs Old: *P* = 0.034. Exon: Old vs Young: *P* = 0.032; Old+OSKM vs Old: *P* = 0.036. Intron: Old vs Young: *P* = 0.045; Old+OSKM vs Old: *P* = 0.071. Intergenic: Old vs Young: *P* = 0.039; Old+OSKM vs Old: *P* = 0.044. NME comparisons: Island: Old vs Young: *P* = 0.023; Old+OSKM vs Old: *P* = 0.025. Shore: Old vs Young: *P* = 0.042; Old+OSKM vs Old: *P* = 0.035. Shelf: Old vs Young: *P* = 0.038; Old+OSKM vs Old: *P* = 0.031. Open Sea: Old vs Young: *P* = 0.041; Old+OSKM vs Old: *P* = 0.029. Promoter: Old vs Young: *P* = 0.029; Old+OSKM vs Old: *P* = 0.028. Gene Body: Old vs Young: *P* = 0.032; Old+OSKM vs Old: *P* = 0.019. Exon: Old vs Young: *P* = 0.028; Old+OSKM vs Old: *P* = 0.022. Intron: Old vs Young: *P* = 0.048; Old+OSKM vs Old: *P* = 0.035. Intergenic: Old vs Young: *P* = 0.038; Old+OSKM vs Old: *P* = 0.029. (**C**). Heatmap showing the proportion of latent DNA methylation components (LMC) across whole skin samples. LMCs and their proportions in samples were generated from decomposition of DNA methylation data by a reference-free cell-type deconvolution approach implemented by the MeDeCom software to estimate cell-type components (Lutsik et al, 2017). Most samples are primarily represented by a single LMC, indicating relatively homogeneous DNA methylation patterns and arguing against substantial cell-type admixture. Aging or rejuvenation treatment is associated with a transition toward a different dominant LMC, reflecting biological shifts in DNA methylation patterns rather than changes in cell-type composition.

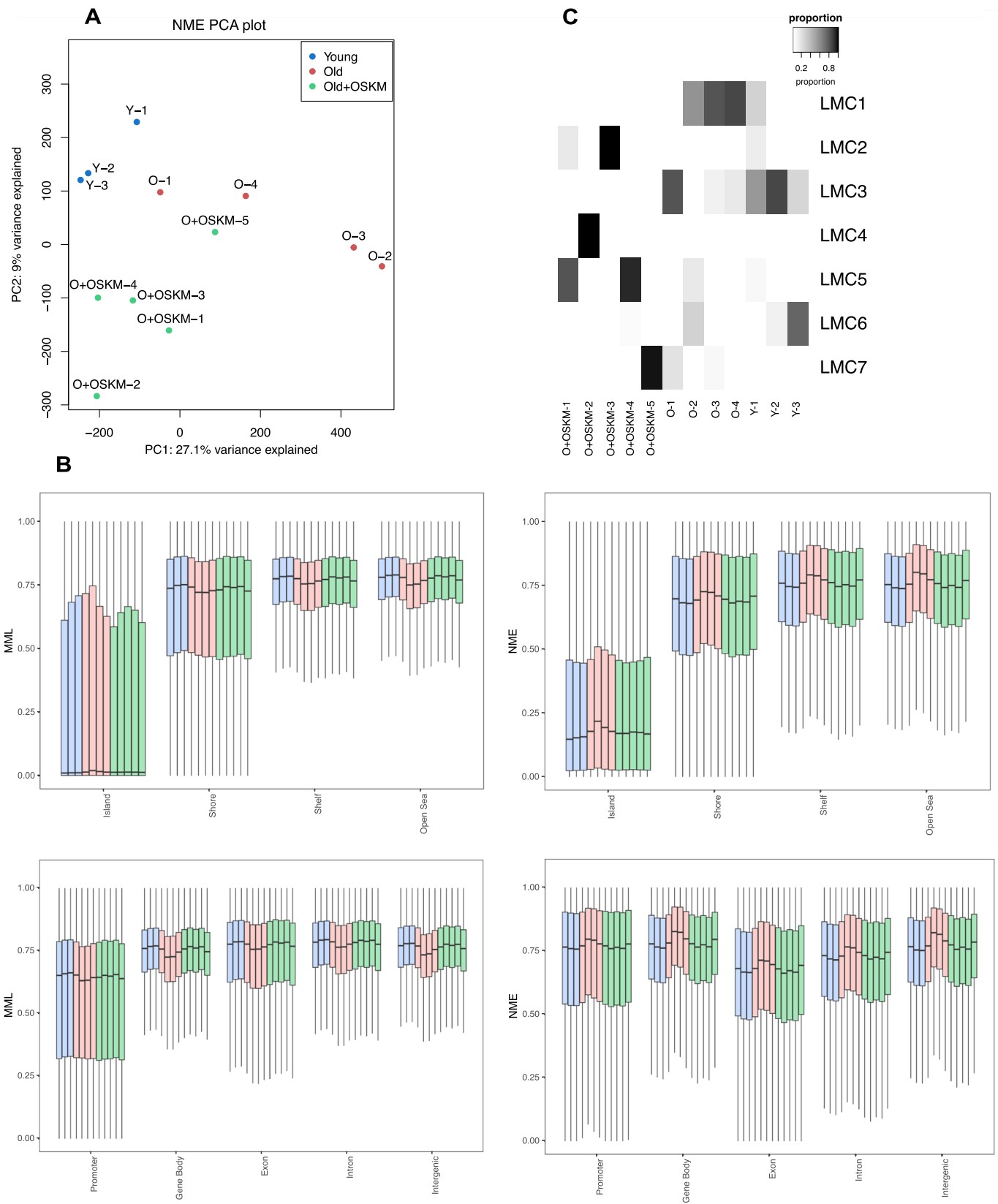

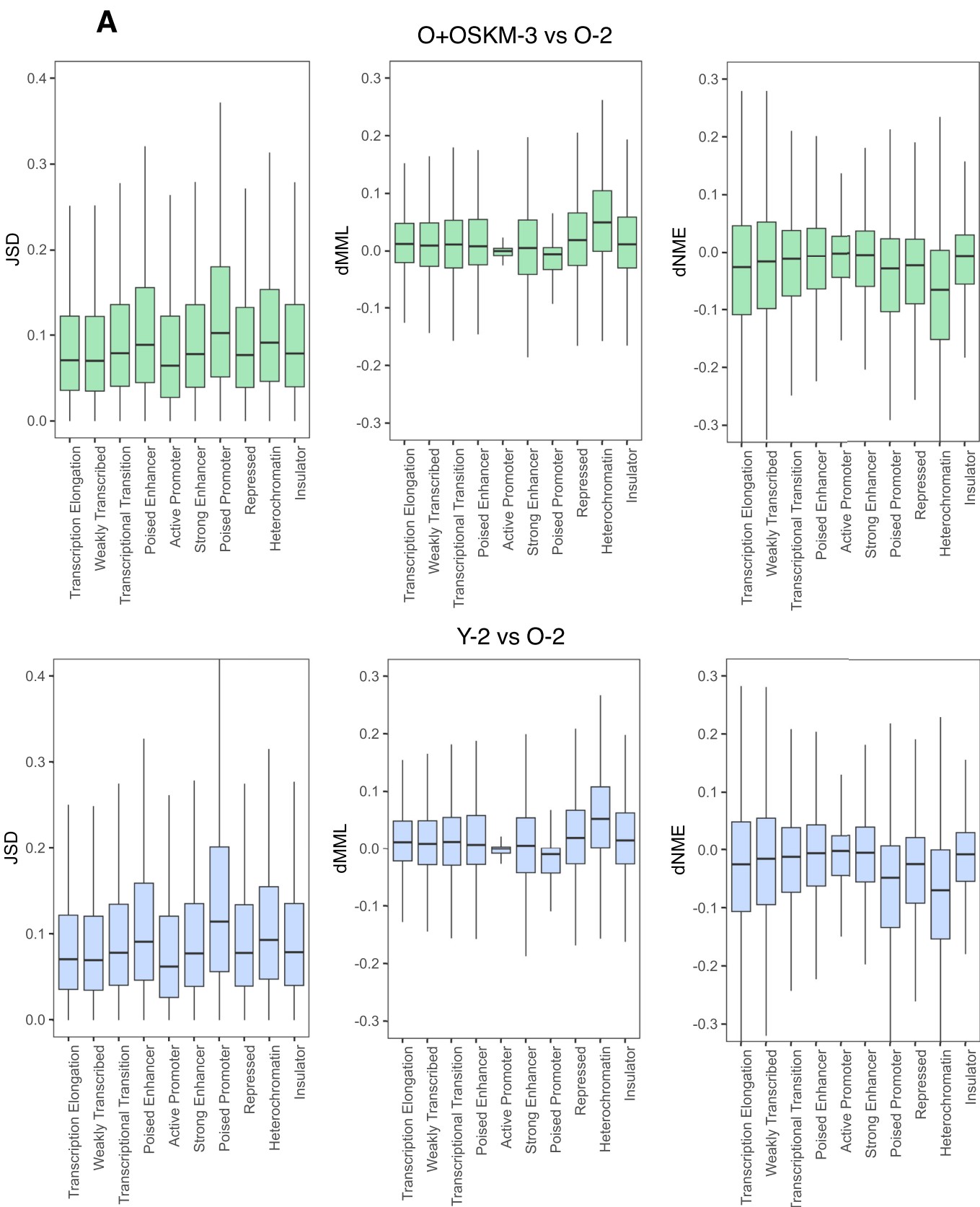

◀ **Figure EV2. DNA methylation changes across chromHMM chromatin states in whole skin during aging and rejuvenation.**

(A) Box plots of dMML, dNME, and JSD distributions across chromatin states defined by chromHMM (derived from mouse embryonic stem cells), based on per-region differences in representative pairwise comparisons between individual whole skin samples, including Young (Y-2), Old untreated (O-2), and Old treated (O + OSKM-3). Each box summarizes the distribution of values across genomic regions assigned to each chromatin state. For each box plot, the central line represents the median, the box bounds correspond to the first (Q1) and third quartiles (Q3), and the whiskers extend to the most extreme data points within 1.5× the interquartile range (IQR) from the quartiles.

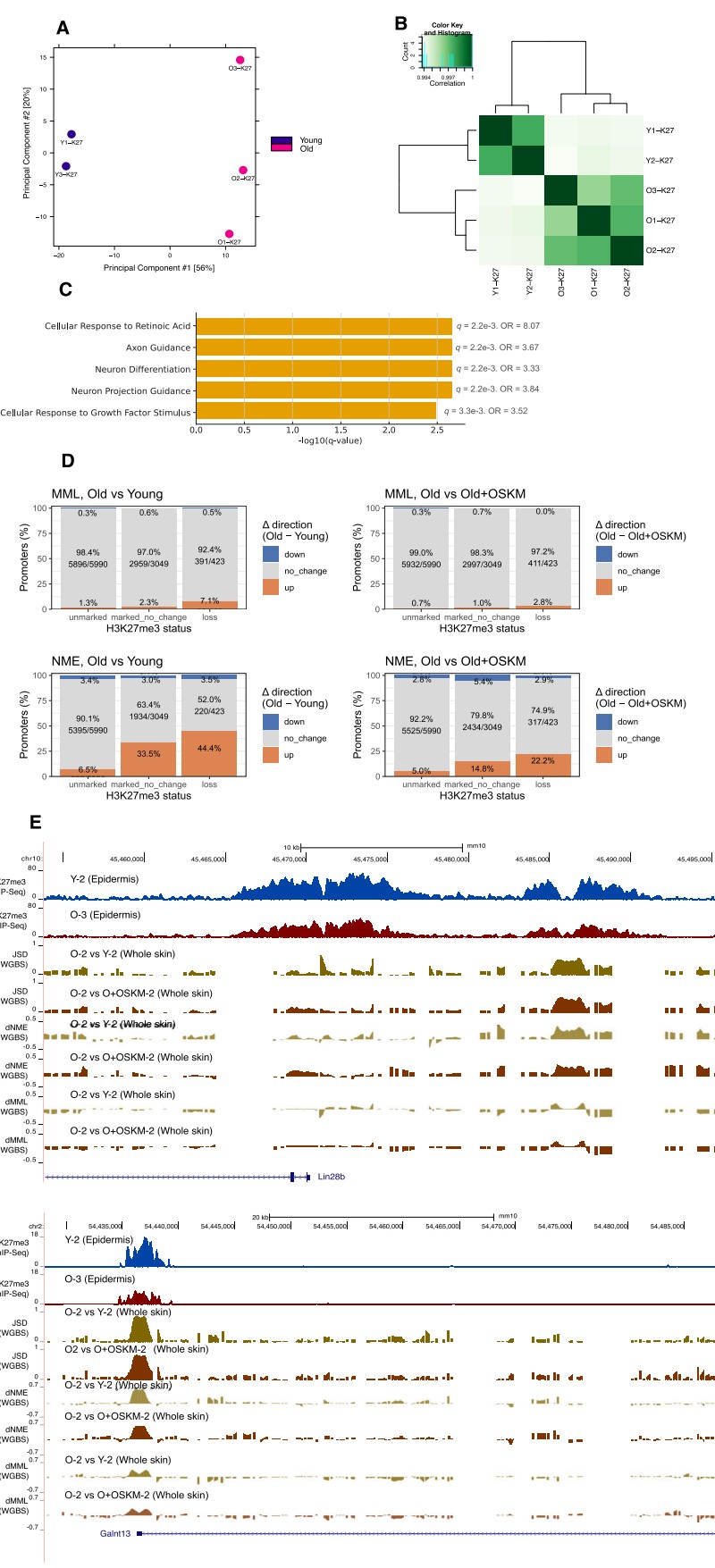

**Figure EV3.   Quality assesment and integration of H3K27me3 ChIP-seq with DNA methylation.**

(A) PCA plot of input-subtracted H3K27me3 genome coverage from young and old tail epidermis samples. (B) Correlation heatmap of input-subtracted H3K27me3 genome coverage from young and old tail epidermis samples. (C) Top five over-represented gene sets from the GO Biological Processes collection identified performing ORA via Enrichr using the genes exhibiting differential H3K27me3 at their promoter in epidermis samples during aging. (D) Barplots showing the proportion of promoters with CpG islands that exhibit increased, decreased, or unchanged MML (top row) and NME (bottom row), measured only within CpG islands, in old untreated (Old) vs Young and Old vs old treated (Old+OSKM) whole skin, stratified by H3K27me3 status (unmarked, marked with no change, or H3K27me3 loss) measured by ChIP on purified epidermis. Values indicate percentages and number of promoters in each category. Promoters were classified as having increased or decreased MML or NME when the difference in condition-level averages exceeded ±5% (see "Methods"). (E) Examples of genes exhibiting loss of H3K27me3 during aging in regions with increased DNA methylation discordance, as indicated by elevated JSD and dNME in Old untreated (Old) samples compared to Young and Old treated (Old+OSKM) samples.

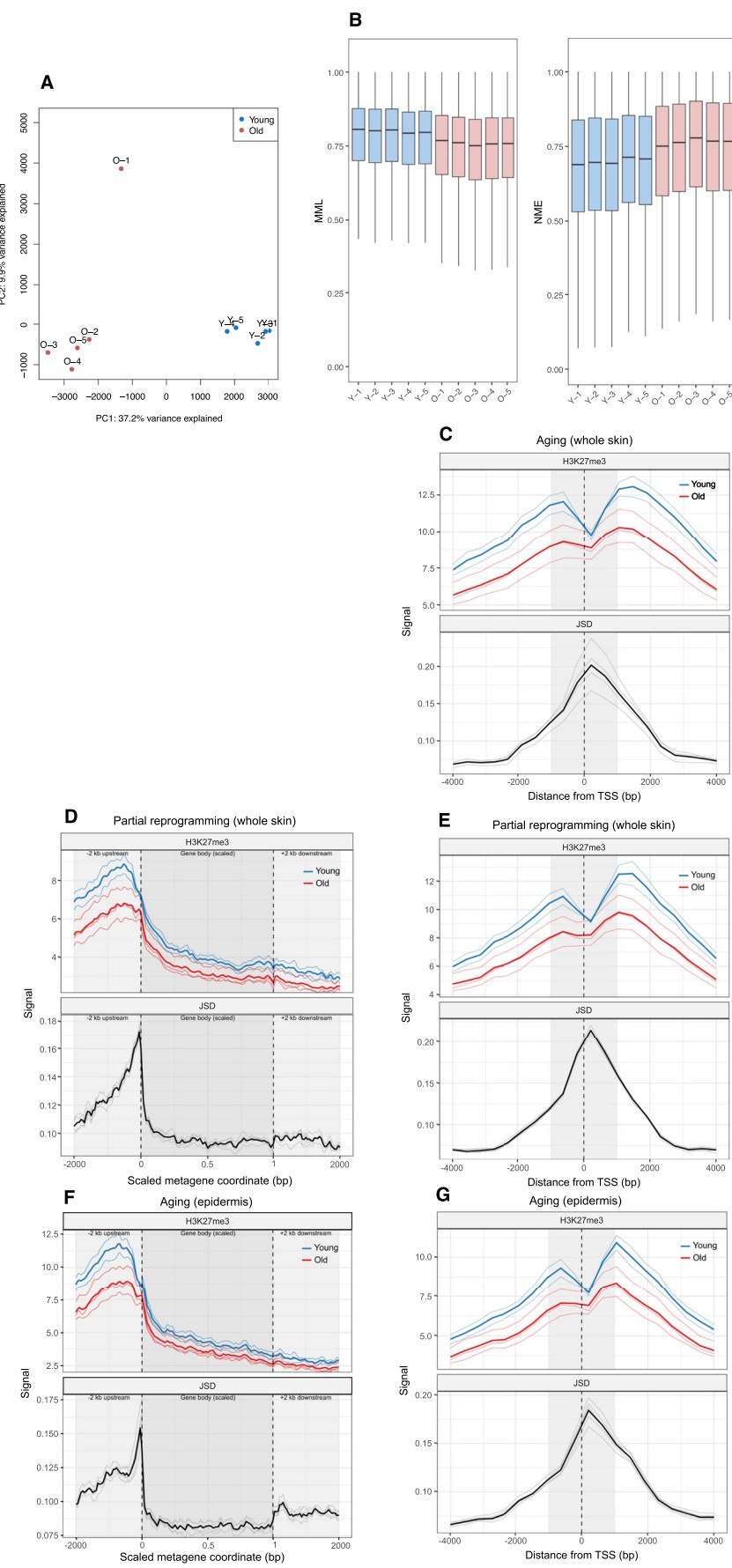

**Figure EV4.  Global DNA methylation patterns in epidermis and metaplot profiles or H3K27me3 occupancy and JSD.**

(A) PCA plot of genome-wide MML in WGBS samples from young and old isolated tail epidermis samples. (B) Box plots of genome-wide MML distributions in WGBS from isolated tail epidermis of young ($n = 5$) and old ($n = 5$) mice. For each box plot, the central line represents the median, the box bounds correspond to the first (Q1) and third quartiles (Q3), and the whiskers extend to the most extreme data points within 1.5× the interquartile range (IQR) from the quartiles. (C–G). Metaplots of H3K27me3 occupancy and DNA methylation discordance (JSD) during aging and rejuvenation. Gene-level plots (D, F) show scaled gene bodies flanked by ±2 kb, while TSS-centered plots (C, E, G) show promoter-proximal regions spanning ± 4 kb around transcription start sites. Thin lines represent individual replicates for H3K27me3 ChIP, while for JSD they represent individual pairwise comparisons, where each old sample is compared against a representative control sample (aging in whole skin: Old vs Y-3; rejuvenation in whole skin: Old+OSKM vs O-2; aging in epidermis: Old vs Y-1). Bold lines represent the average signal. (C) TSS-centered metaplot for the set of genes with significant promoter-restricted JSD-based methylation discordance during aging in whole skin. (D) Gene-level metaplot for the set of genes with significant JSD-based methylation discordance over promoters and gene bodies in partial reprogramming in whole skin. (E) TSS-centered metaplot for the set of genes with significant promoter-restricted JSD-based methylation discordance in partial reprogramming in whole skin. (F) Gene-level metaplot for the set of the top 1000 JSD-ranked genes (based on discordance over promoter and gene body) during aging in purified epidermis. (G) TSS-centered metaplot for the set of the top 1000 JSD-ranked genes (based on discordance over promoter) during aging in purified epidermis.

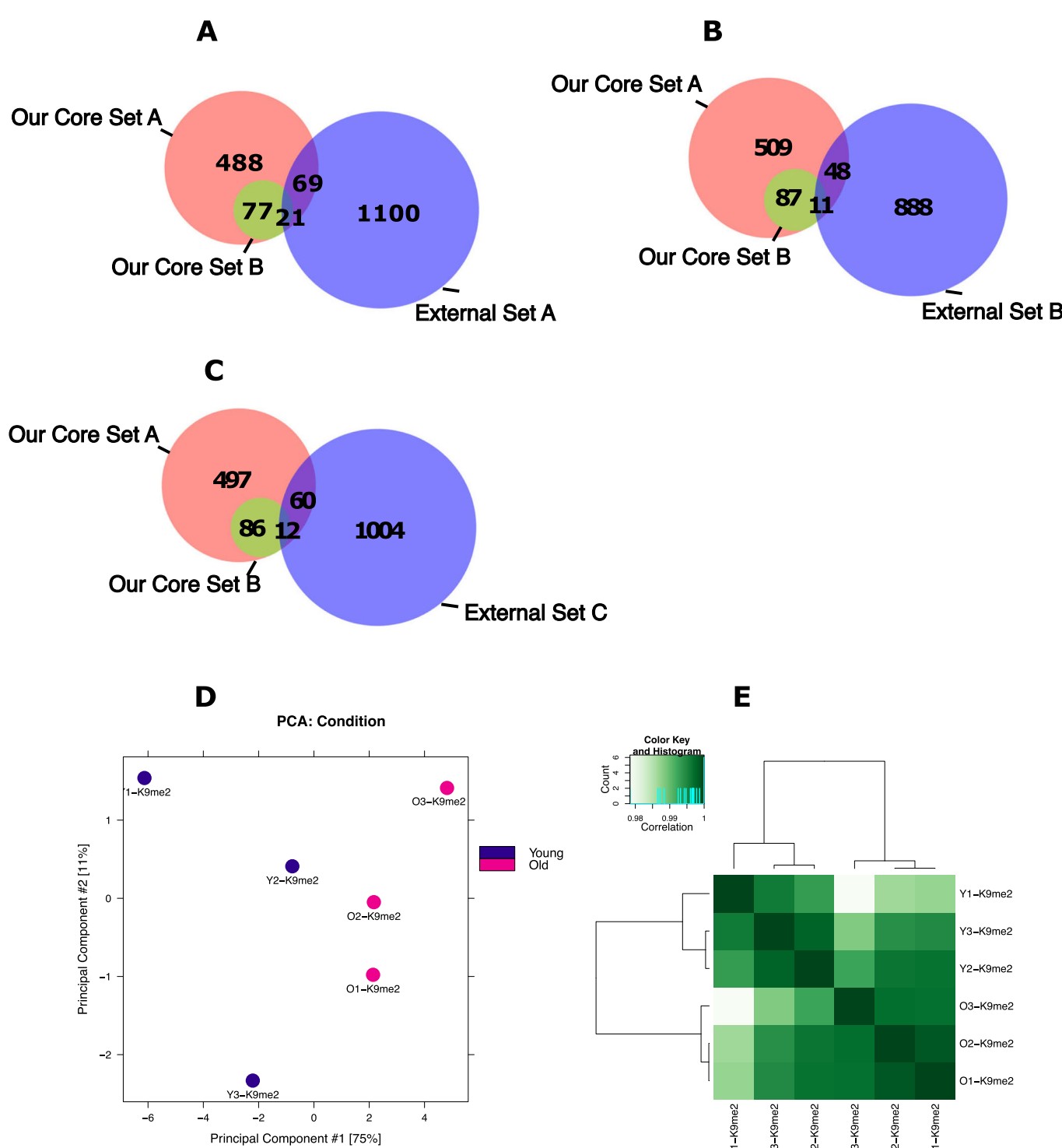

**Figure EV5.    Cross-study overlap of age-related epigenetically dysregulated genes and quality assessment of H3K9me2 ChIP-seq.**

(A–C) Overlap between core sets of epigenetically dysregulated genes identified in this work and gene lists from other studies. 'Our Core Set A' refers to genes with DNA methylation discordance (significant JSD over promoters and gene body) in both aging and partial reprogramming-mediated rejuvenation, identified in our WGBS data. 'Our Core Set B' refers to genes included in 'Core set A' in addition to significant differences in H3K27me3 enrichment during aging, identified in our ChIP-Seq data. (A) Overlap with External Set A (Horvath et al, 2022a). The list is based on genes linked to CpGs significantly associated with aging in naked mole rats. (B) Overlap with External Set B (Moqri et al, 2024). The list is based on genes linked to 'PRC2-enriched lowly methylated regions', as defined by the authors, in human epidermis. The authors claim 90% of such regions gain DNA methylation during aging. (C) External Set C (Yang et al, 2023b). The list is based on genes linked to significant changes in H3K27me3 enrichment during liver aging. (D) PCA plot of input-subtracted H3K9me2 genome coverage from young and old tail epidermis samples. (E) Correlation heatmap of input-subtracted H3K9me2 genome coverage from young and old tail epidermis samples.

