## [Peer Review File · Molecular Systems Biology]

Convergence of aging- and rejuvenation-related epigenetic alterations on PRC2 targets

Oscar Camacho, Michael Koldobskiy, Pradeep Reddy, Atharv Oak, Yuxiang Sun, Kenna Sherman, Juan Carlos Izpisua Belmonte, and Andrew Feinberg

Corresponding author(s): Andrew Feinberg (afeinberg@jhu.edu), Juan Carlos Izpisua Belmonte (jcbelmonte@altoslabs.com), Michael Koldobskiy (mak@jhmi.edu)

Review Timeline:

Transfer Date:	25th Feb 25
Editorial Decision:	1st Apr 25
Revision Received:	11th Dec 25
Editorial Decision:	16th Jan 26
Revision Received:	20th Jan 26
Accepted:	21st Jan 26

Editor: Jingyi Hou

Transaction Report: The first round of review of this manuscript was performed at another journal.

We thank the reviewers for their thoughtful comments and suggestions on improving the manuscript. We respond to all reviewer comments below and have made changes to the manuscript based on these suggestions. It has taken quite a bit of time to complete the suggested ChIP-seq experiments – in fact, no one had ever done H3K27me3 native ChIP in primary mouse epidermis in physiologic aging, and this required development of new protocols – but we agree that this work has improved the paper substantially, providing a biological context and mechanism for our DNA methylation findings. Additional novelty to this work comes from its comprehensive, unbiased analysis of all CpG sites in the genome (in contrast to custom arrays of a limited number of sites that had been used to develop aging clocks), the incorporation of epigenetic stochasticity (quantified as DNA methylation entropy) – which is known to be incredibly important in aging but has not received a quantitative, and the finding of PRC2 regulation of both aging and OSKM rejuvenation.

Reviewer #1:

Remarks to the Author:

In this manuscript, Koldobskiy et al performed whole genome bisulfite sequencing (WGBS) on skin isolated from mice undergoing long-term partial reprogramming (via OSKM expression) along with young and old controls. The WGBS data provided a quantitative estimation of the methylation status of ~15 million CpGs at base pair resolution. The reprogrammed mice (n=5) were under dox treatment from 15-22 months of age, while old untreated controls were 22 months old (n=4) and young untreated controls were 3 months old (n=3). The reprogramming paradigm showed evidence of histological and functional rejuvenation of the skin (published data). Data analysis captured both mean methylation levels (MML) as well as normalized methylation entropy (NME) with the identification of genomic regions with DNA methylation discordance using the Jensen-Shannon distance (JSD). While the involvement of PRC2 in aging/longevity has been implicated in multiple studies to date, that PRC2 targets show reversible changes during reprogramming is new.

We appreciate the reviewer's summary and crystallization of the novelty of the paper implicating PRC2 targets in reversible changes of aging.

The following points need to be addressed:

1. Several recent publications have implicated PRC2 in longevity and should be discussed and cited. These include <https://www.biorxiv.org/content/10.1101/2022.06.03.494609v1>, PMID: 35368774, PMID: 37116496, PMID: 36638792 and PMID: 35580182.

We thank the reviewer for bringing up these relevant references and we have incorporated all into the text. To briefly contrast the contributions of these papers to our current work:

The BioRxiv paper now published as Moqri et al, *Nature Comms* (PMID 39009581) re-analyzes multiple published aging datasets and compares them to PRC2 sites identified in embryonic stem cells (not the aging cells themselves). Importantly, Moqri cites our manuscript (which had also been uploaded to BioRxiv at the time of its initial submission) as reference #54, and they say explicitly in the text that our finding shows the biological significance of altered methylation at PRC2 regions in aging. The Horvath papers (PMIDs 35368774 and 35580182) use custom

mammalian methylation arrays encompassing ~37,000 probes to follow aging-associated changes in naked mole rats and dogs. There is an enrichment of H3K27me3 marks, EED binding sites and PRC2 targets among small numbers of genes that are proximal to hypermethylated age-related CpGs. In contrast to our work, these papers do not evaluate rejuvenation, and do not investigate epigenetic changes in specific tissues, as the PRC2 gene sets used for enrichment are derived from other cell types, mostly embryonic stem cells. Further, we evaluate all CpGs genome-wide, capture methylation stochasticity, and relate these changes to H3K27me3 alterations assays by ChIP-seq in aging skin. Yang et al (PMID 37116496) investigates quiescent non-dividing liver (unlike aging skin we study which does divide) and hepatectomy-induced regeneration at the level of chromatin, but not rejuvenation in aging as we do, and not studying methylation, let alone entropy. We also added a new Supplementary Figure 5 showing that the genes we identify are largely distinct from those other studies, as expected for the reasons stated above. Yang (PMID 36638792) shows that repair of DNA damage in cultured fibroblasts increases H3K27ac and decreases H3K27me3 in genes related to disruption of cell identity, and the authors speculate a relationship to epigenetic entropy. We end the paper with citation to this paper, as it follows in a lovely way a brief discussion of the information-theoretic model of aging we proposed in Ref. 9.

All of these papers are interesting and well worth citing but do not “scoop” the heart of the matter in our manuscript, of methylation entropy in aging and rejuvenation, its enrichment for PRC2, and its overlap with H3K27Me3 (now confirmed thanks to the reviewers’ suggestions). Please note that in the text we mention these works but do not point out the limitations, just contrast them with the present study.

2. It is unclear in Fig. 2b, how much of an overlap there is between EZH2 binding sites, H3K27me3-enriched regions, and poised promoters. They are likely to have a large overlap. Could the authors show this in a Venn diagram? That these enrichment sites are derived from embryonic stem cells should be clearly stated in the text with reference. What happens to PRC2 binding or H3K27me3 in differentiated skin cells in young, during aging and reprogramming? Could the authors speculate?

Consistent with previous literature (PMID 30951652), using chromatin annotation tracks and ChIP-seq data from mouse ESCs, we show a large overlap between EZH2 binding sites, H3K27me3-enriched regions and poised promoters in mESC. 81.6 % of EZH2 binding sites and 71.5% of poised promoters overlap with H3K27me3-enriched regions. At the same time, 78.3% of EZH2 binding sites overlap with poised promoters. However, as observed in other reports (PMID 24289921) the H3K27me3 mark is also found in other genomic regions. Given that creating a Venn diagram based on overlaps of genomic regions can be misleading because multiple regions from one group can overlap to a single region from another and there is no 1:1 correspondence, we associated the different genomic regions to overlapping promoters and generated a Venn diagram based on the shared corresponding genes. We included the Venn diagram in Supplementary Figure 2. We updated the text and the figures to clearly indicate these genomic regions are derived from mouse embryonic stem cells, referencing the source in the section ‘Data and code availability’.

As part of the revisions to this manuscript, we performed native ChIP-Seq to study how the H3K27me3 histone mark changes during aging and identified a wide-spread, genome-wide loss of this mark. Those sites with loss of H3K27me3 in old skin significantly overlap with the highly epigenetically discordant genes we identified with DNA methylation analysis. We speculate that PRC2 binding is also diminished during aging and that reprogramming would restore both PRC2 binding and H3K27me3 levels, rejuvenating the epigenome in a similar manner as we have shown with DNA methylation in this study.

3. Could the authors also speculate on the biological meaning of the finding that poised promoters show the greatest JSD?

Our prior information-theoretic analysis of DNA methylation showed that regions characterized by the highest entropic sensitivity represented deformable potential energy landscapes, likely corresponding to differentiation branch points (PMID 28346445). After differentiation, regions associated with pluripotency/lineage plasticity have less need for maintenance of epigenetic fidelity, and may be more prone to age-associated epigenetic drift / erosion of epigenetic information with aging. We speculate that the enrichment of bivalent/poised regions among sites of greatest methylation change (JSD) represents the inappropriate opening of chromatin regions in long-lived organisms. We have added the requested language at the end of the manuscript.

3. The background of the mice (C57BL/6) should be revised to denote substrain/provenance – C57BL/6J or C57BL/6N, C57BL/6NCrl, or C57BL/6JN. Please also provide sex information.

All the mice used in the study were in a C57BL/6J background, with a balanced distribution of sexes in each group. We have updated this information in the text and included it as sample metadata in Supplementary Table S1.

4. What does the “-2” represent in the figures? If it represents only the 2nd replicate, it is advised to show the “mean” methylation profiles/GSEA output rather than a single sample (for example in Fig. 1c-e, Fig. 2a, c and Fig. 3).

We have clarified in the text when pooled/group profiles are used and when specific pairwise comparisons are employed. Individual comparisons are useful to highlight DNA methylation entropy as an encapsulation of methylation stochasticity within the sample (across multiple reads from a single sample), rather than inter-sample variability. The statistical justification for this approach is describe in detail in Jenkinson et al Nature Genetics (PMID 28346445, Supplementary note, section 2.3).

5. All code for methylation analysis should be deposited to a public code repository.

All code used in the analysis has been deposited on GitHub (https://github.com/ocamach1/PRC2_aging_rejuvenation). Source code for InformME as utilized in this manuscript is also available on GitHub <https://github.com/GarrettJenkinson/informME> . We have included this information in the text under the section ‘Data and code availability’.

Reviewer #2:

Remarks to the Author:

In this manuscript, the authors investigated the change of DNA methylation level in skin from mice subjected to partial reprogramming (rejuvenated) as well as young and aged controls using whole genome bisulfite sequencing. Authors found that gain of the DNA methylation in the aged samples when compared to other samples is highly associated with the enrichment in Polycomb repressive complex 2 (PRC2) gene targets as shown by enrichment in target of EZH2, SUZ12 etc. and genes harboring H3K27me3 mark. Finally, gene expression analysis suggested PRC2 drives age-related gene expression changes.

To our understanding, author's conclusion is based only on differential patterns of epigenetic marks among the experimental groups. Most importantly, we would say that this study has not advanced in terms of the concept that aging-associated increase in DNA methylation in PRC2 target regions as authors discussed at Line 114-116. In addition, authors have not performed any experiments related to PRC2 dynamics such as ChIP-seq during aging. Therefore, we have to say that it is hard to claim the importance of PRC2 as a marker of aging based solely on this observation at this moment.

As suggested by the reviewer, based on the striking enrichment for H3K27me3-marked genes among differentially methylated genes in both aging and rejuvenation, we sought to directly examine H3K27me3 dynamics in aging. To this end, we developed and implemented a novel native ChIP-Seq protocol to profile H3K27me3 binding in isolated young and old epidermis from tail skin. To our knowledge, this is the first time an epigenetics assay profiling chromatin modifications (i.e. ChIP-Seq, ATAC-Seq) has been performed on primary mouse skin during aging. Due to the inherent challenges of processing adult mouse skin, we developed specific methods to dissociate epidermis and isolate nuclei from keratinocytes and performed ChIP in native, non-crosslinked chromatin, because of the low yields in immunoprecipitated DNA and the low signal-to-noise ratio that is associated with conventional crosslinked ChIP-Seq. We purified the epidermal layer to minimize potential biases related to cell-type composition. Exogenous *Drosophila* chromatin was spiked in before immunoprecipitation to enable quantitative comparisons (ChIP-Rx). These results comprise a major new part of the manuscript, presented as Figure 3. Importantly, differential peak analysis revealed a widespread loss of H3K27me3 across the genome (Figure 3a), with a significant reduction in 43% of peaks, a substantial proportion of which were located at gene promoters (Figure 3b). We integrated our ChIP-Seq data with DNA methylation analysis and found a significant overlap between genes displaying differential H3K27me3 peaks over their promoters by native ChIP between young and old in isolated epidermis and differentially DNA methylated genes (significant JSD over the promoter) with aging and partial reprogramming in dorsal skin. We aimed to identify a core set of genes identified by our multimodal analysis as those with significant methylation discordance during aging, significant methylation discordance during partial reprogramming-induced rejuvenation, and significant change in H3K27me3 distribution during aging. This produced an overlap set of 98 genes, representing a core set of PRC2 regulated genes that are epigenetically altered during aging and rejuvenation. Prominent examples within this core set include *Wnt5a*, *Lin28b* and *Sox11*.

Reviewer #3:

Remarks to the Author:

Koldobskiy and colleagues performed whole genome bisulfite sequencing in mouse fibroblasts of young, old and rejuvenated animals to study the epigenetic mechanisms associated with organismal aging and rejuvenation, observing an enrichment in PRC2 targets and entropy.

General comments

1. Novelty: The authors report that aging is associated with hypermethylation of regions repressed by the PRC2 complex and increased DNA methylation entropy. The level of novelty of these findings is low, as they are well-known observations in the field (e.g. PMID: 27654999, and reviewed years ago by PMID 25637097, among others). Upon rejuvenation induced by the Yamanaka factors, this hypermethylation is decreased. Is this not the expected observation? Perhaps it is a subjective assessment, but for a top article in aging epigenetics, I was expecting more mechanistic insight.

We have added further clarification to the text on the differences between our approach and prior work implicating PRC2 alterations in aging (as also discussed in the response to reviewer 1, point 1). Our study involves a comprehensive, unbiased analysis of DNA methylation and methylation entropy with aging as well as rejuvenation. This differs from prior array-based studies of whole-blood mean methylation in aging, particularly in the capacity to find entropic changes with aging. The finding of a reversible, entropic age-associated epigenetic signature is highly novel. The fact that prior studies looking only at mean methylation and at small numbers of CpGs sites also suggest a relationship to PRC2 highlights the importance and generality of this mechanism. We also now show that we are identifying a novel set of core genes involving aging, rejuvenation, DNA methylation, and H3K27Me3. As suggested, we have now carried out additional work including H3K27me3 ChIP-seq in this aging system to further investigate the underlying mechanism. We also added a reference to that wonderful opinion article from 2015 that I think draws the first connection between cancer/aging/hypermethylation/PRC2.

2. Study design: The manuscript would have gained in impact several tissues had been analyzed.

The ChIP experiments on purified epidermis requested by Reviewer 2 were highly valuable but also very labor intensive, and we see it as a strength that we have probed in depth the epigenome in rejuvenation in primary experiments in skin. It would not have been practical for us to begin anew on another tissue in a reasonable time frame.

3. Analysis: a very important factor is that authors do not tackle aging-associated cell-composition issues, which is indeed a confounding variable in this type of studies.

We agree that variation in cell-composition is a potential confounding variable in epigenetic studies during aging. However, originally we could not address this issue experimentally due to the inherent challenges of studying skin tissue. Separation of the epidermal and dermal fractions in dorsal skin is only feasible in newborn mice, during the telogen or resting phase of the hair cycle, in which the hair follicle is dormant, and growth of the hair shaft does not occur. Dissociation of dorsal skin into single cells for subsequent sorting is also challenging, yielding

low numbers of cells and diminished viability. This limitation is well recognized in the aging field and the vast majority of publications studying epigenetic changes in skin aging are conducted using whole dorsal skin or in vitro models such as cultured fibroblasts. Nevertheless, we have addressed this issue experimentally, by performing ChIP-Seq and WGBS on purified epidermis from skin tail in young and old mice that matched the ages of the original cohorts. We decided to purify epidermis to tackle the potential aging-associated cell-composition issues. Unlike dorsal skin, the separation of epidermal and dermal fractions by dispase treatment is possible at any age of the mice due to a lower density of hair follicles, making it a good model to study skin aging. To our surprise, we could only find a very limited number of publications based on a chromatin assay (i.e., ChIP-Seq, ATAC-Seq) in primary mouse skin (PMIDs 32314219, 36708949), all of which were performed on purified skin fractions in whole dorsal skin from newborns. To our knowledge, this is the first study to examine chromatin changes in skin during aging. The age-related DNA methylation changes identified in whole dorsal skin significantly overlap with those found in isolated tail epidermis, supporting the validity of the original data.

Specific comments

4. I cannot find any detailed explanation on sample preparation. How was the skin processed? Do the samples have the same proportion of cell types? In line with my previous general comment, as each cell type shows a specific DNA methylation, it could be that the observed differences are related to different cellular composition across samples. Some deconvolution algorithm may help.

We updated the Methods section to include a brief description of dorsal skin collection and the procedure for isolating epidermis from tail skin. Dorsal skin was collected using surgical scissors and immediately stored in liquid nitrogen without additional processing until preparation for DNA methylation and ChIP-Seq analyses. These samples should contain both the dermal layer (primarily composed of fibroblasts) and the epidermal layer (primarily composed of keratinocytes) (PMID 16443746). According to previous literature (PMID 32109378), the cellular composition of murine dorsal skin consists of approximately 45% fibroblasts, 30% keratinocytes, and 10% endothelial cells. The epidermis is mostly (95%) composed of keratinocytes (PMID 24497224), making it an ideal tissue to study epigenetic changes during aging mitigating the risk for potential aging-associated cell-composition bias. We did not apply a deconvolution algorithm in our dorsal skin samples.

5. With regard to figure 1:

a. How is the PCA done: with genomic bins? With mean DNA methylation? Could you do a PCA of both MML and NME ?

Our pipeline to analyze DNA methylation (informME) generates mean methylation and normalized entropy values for non-overlapping genomic bins of 150 bp of size. The PCA plot is based on MML taking into account all the 150 bp genomic bins in the genome. We generated a PCA plot using the average DNA methylation level per CpG site and the result was undistinguishable when compared to the PCA plot generated based on genomic bins. We included a PCA plot based on NME in Supplementary Figure S1.

b. What's the genome-wide correlation between MML and NME?

The genome-wide correlation between MML and NME in the samples from this study is -0.32 on average. We describe in detail the relationship between MML and NME in our publication where we present the methods (PMID 28346445). Genome-wide scatter plots of MML and NME are shown in Supplementary Figure S2 of this publication. In summary, MML is not predictive of NME. In this line, the DNA methylation landscape change for a large proportion of regions can only be detected by differences in NME but not in MML (PMID: 28346445, 36762477).

c. Fig 1b, is there any difference genome-wide in other chr context? How was chhm defined?

As shown in Supplementary Figure S2, when exploring the differences in DNA methylation in regions defined with chromHMM, we identify that poised promoters are the only regions that gain MML (and also NME) during aging. However, the regions defined as heterochromatin are the ones showing the largest differences in MML, with a pronounced reduction in this metric and an increase in NME. The old treated samples show a reversal of those changes in those regions in both MML and NME. Annotations for regions defined by chromHMM were obtained from the ENCODE project and are generated based on enrichments in the occupancy of different histone marks across the genome. The maps of histone marks used are also available from the ENCODE project and were generated from ChIP-Seq experiments. According to chromHMM, poised promoters are promoter regions enriched in both H3K27me3 and H3K4me3, while heterochromatin is defined by regions enriched in H3K9me3.

d. Fig. 1b seems to be redundant with Fig. 1c-e. Also, they arbitrarily choose the most different or similar samples to plot

In Fig. 1c-e we wanted to emphasize that the OSKM treatment globally restores MML and NME to a state that resembles the young samples. We recognize they can seem redundant as global trends are already shown in Fig. 2 in the form of a histogram, including all samples.

e. The group colors in Fig. 1c do not match those from Fig. 1b. May it be that the labels in Fig. 1c are swapped?

We thank you for bringing this error to our attention. The labels in Fig. 1c were swapped and we corrected the figure.

f. Fig. 1c, all cases could be represented, as few are shown.

Even though we only show a comparison between representative samples in the density plots, all the samples are represented in the histograms of Fig. 1b and the subsequent analyses and statistics for differential DNA methylation take into account all the available samples from the different groups.

6. With regard to figure S1:

a. Correct me if I am wrong, but this figure does not show the same trend as Fig. 1c in islands, for MML. Please explain.

Most CpG islands remain unmethylated in somatic cells (PMID 22641018). In Figure S1 we show CpG islands remain highly unmethylated during aging, with a very mild increase in MML in old compared to young samples. Indeed, this differs from the trend of global age-related hypomethylation we observe at the genome-wide level and reflects the majority of CpG islands may not show significant changes in MML during aging. As this is a summary plot describing a general tendency in such regions, it does not preclude that there are specific islands with significant changes in MML during aging. Although CpG islands in intragenic and gene body regions can have tissue-specific patterns of methylation and are prompt to change upon environmental exposures, CpG islands associated with transcription start sites rarely show tissue-specific methylation patterns and are less plastic (PMID 20613842). Instead, CpG shores, located as far as 2 kb from CpG islands, have highly conserved patterns of tissue-specific methylation (PMID 19151715) and we show MML in those are more prone to change during aging.

b. Could you please distinguish promoters from gene body, exon, intron, intergenic regions?

We define promoters as regions that are 2000 bp upstream and downstream from the transcriptional start site of the gene and have clarified that in the section on genomic features and annotations. Gene bodies do not include the first 2000 bp in the 5' of the gene that overlap with the defined promoter. In the same manner, intergenic regions do not include the 2000 bp upstream of the transcriptional start of a gene, not overlapping with defined promoters.

c. Does “gene body” encapsulate both exons and introns?

Yes, it encapsulates both exons and introns.

7. With regard to your statement: “We identified the most epigenetically discordant genes and evaluated for enrichments among the top 500 genes in each group-wise comparison using Gene Set Enrichment Analysis (GSEA).”

a. The authors identify top 500 genes by methylation. But, then, how is the GSEA done? Using a measure from the JSD? What's the background/universe for the enrichment analysis?

As described in the Methods section, when performing comparisons between samples, we calculate JSD values for every non-overlapping 150 bp genomic bin in the genome. To have a representative JSD value for a promoter or gene body of each gene, we calculate the square root of the average of the squared JSD values within all analysis 150 bp bins that overlap with the promoter or gene body. As shown in Supplementary Tables S2, S3 and S7, we performed our analysis considering only promoters and also combining both promoters and gene bodies. In the Methods section we describe how we implement hypothesis testing to decide if a promoter or gene body is statistically different among 2 conditions based on JSD, generating a list of genes

with adjusted p-values associated to the statistical differences that the corresponding promoter or gene body shows. We selected the top 500 genes that showed the most significant differences based on adjusted p-value. The GSEA was done using the software from the Broad Institute (PMID 16199517), which performs GSEA using a hypergeometric test and provides a p-adjusted value and an odds ratio. The background for the enrichment analysis was set to be all the genes in the mouse genome according to the GENCODE VM23 annotation, used by the TxDb.Mmusculus.UCSC.mm10.knownGene package in R, as that it is the number of genes considered in the DNA methylation differential analysis. We updated the Methods section to include a better description of the analyses performed.

b. I am surprised to see that the comparisons are 1 vs 1 single samples. Is this right? Why not performing statistical comparisons using all available samples?

JSD values are computed in 1 vs 1 single samples. However, when performing hypothesis testing, we use all the available samples. First, we generate a null distribution based on JSD derived from comparisons among the control samples (JSD values from control 1 vs control 2, control 2 vs control 3, control 1 vs control 3). Then, we use the calculated JSD derived from all the possible test vs control comparisons (i.e. JSD values from test 1 vs control 1, test 2 vs control 1, test 2 vs control 1, test 2 vs control 2...) and perform hypothesis testing against the null hypothesis based on the null distribution. This method was previously published in Nature Genetics and BMC Bioinformatics (PMIDs 28346445, 29514626, 30961526) and we have included these references.

8. It seems there is no directionality of DNA methylation in the JSD comparisons. However, directionality is very relevant in DNAm regulation.

The JSD is a symmetric metric that quantifies differences in probability distributions of, in this case, DNA methylation patterns between two samples (PMID 28346445). It can be driven by differences in MML or NME and its value does not include directionality. We agree directionality of the DNA methylation changes is very relevant for epigenetic regulation. We provide additional tests and plots to explore the direction of the differences in MML and NME in the manuscript.

9. With regard to the methods section;

a. Authors use informME to analyze DNAm data: What is, conceptually, or biologically, the concept behind the differences between the regions? What is the expected effect size?

InformME is an alternative to methods like BSmooth, but includes the ability to discover differences in probability distributions other than mean, such as entropy. JSD takes a value from 0 to 1. We have added the JSD values to Supp. Table S2,3. For q-values <0.05, the JSD in this dataset is typically >0.25.

b. Please provide quotations for Bismark, Samtools, Mouse Encode, DESeq2, etc.

We updated the Methods section including citations for the software and R packages used in the analyses.

10. For Boxplots in general, what are the asterisks between the groups? To which test do they refer to?

Differences in global distributions of MML and NME were statistically tested using the Mann-Whitney U test. Significant differences were indicated in histogram plots with an asterisk. We updated the Methods section including this information.

1st Apr 2025

Manuscript Number: MSB-2025-12937-T

Title: Convergence of aging- and rejuvenation-related epigenetic alterations on PRC2 targets

Author: Andrew P. Feinberg

Oscar Camacho

Michael Koldobskiy

Pradeep Reddy

Atharv Oak

Yuxiang Sun

Juan Carlos Izpisua Belmonte

Dear Andy,

Thank you for submitting your manuscript to Molecular Systems Biology. We have now received the enclosed reports from the two reviewers who were asked to assess your work and your response to the reviewers' comments from the previous journal. As you will see, both Reviewer #1 (who reviewed the previous version of the study) and Reviewer #2 (who is new to this manuscript) have raised a number of concerns that should be carefully addressed in a revision of the current manuscript.

The reviewers' recommendations are relatively clear, so there is no need to reiterate the points listed below. All the issues raised by the reviewers need to be satisfactorily addressed. During our pre-decision cross-commenting process (in which the reviewers are given a chance to make additional comments, including on each other's reports), Reviewer #2 agreed with Reviewer #1's concerns regarding code availability and the selection of Figures 1c-e. These comments are included below the reviewers' reports.

As you may already know, our editorial policy allows in principle a single round of major revision, and it is therefore essential to provide responses to the reviewers' comments that are as complete as possible. Please feel free to contact me in case you would like to discuss in further detail any of the issues raised by the reviewers.

On a more editorial level, we would ask you to address the following issues:

- Please provide a .docx formatted version of the manuscript text (including legends for main figures, EV figures and tables). Please make sure that the changes are highlighted to be clearly visible.
- Please provide individual production quality figure files as .eps, .tif, .jpg (one file per figure).
- Please provide a .docx formatted letter INCLUDING the reviewers' reports and your detailed point-by-point responses to their comments. As part of the EMBO Press transparent editorial process, the point-by-point response is part of the Review Process File (RPF), which will be published alongside your paper.
- Please note that all corresponding authors are required to supply an ORCID ID for their name upon submission of a revised manuscript.
- We replaced Supplementary Information with Expanded View (EV) Figures and Tables that are collapsible/expandable online (see examples in <http://msb.embopress.org/content/11/6/812>). A maximum of 5 EV Figures can be typeset. EV Figures should be cited as "Figure EV1, Figure EV2" etc... in the text and their respective legends should be included in the main text after the legends of regular figures.

Additional Tables/Datasets should be labeled and referred to as Table EV1, Dataset EV1, etc. Legends have to be provided in a separate tab in case of .xls files. Alternatively, the legend can be supplied as a separate text file (README) and zipped together with the Table/Dataset file.

For the figures and tables that you do NOT wish to display as Expanded View figures, they should be bundled together with their legends in a single PDF file called *Appendix*, which should start with a short Table of Content. Each legend should be below the corresponding Figure/Table in the Appendix. Appendix figures and tables should be referred to in the main text as: "Appendix Figure S1, Appendix Figure S2, Appendix Table S1" etc. See detailed instructions regarding expanded view here: <https://www.embopress.org/page/journal/17444292/authorguide#expandedview>.

- Before submitting your revision, primary datasets (and computer code, where appropriate) produced in this study need to be deposited in an appropriate public database (see <http://msb.embopress.org/authorguide-dataavailability> <https://www.embopress.org/page/journal/17444292/authorguide#dataavailability>). Please remember to provide a reviewer password if the datasets are not yet public. The accession numbers and database should be listed in a formal "Data Availability" section (placed after Materials & Method) that follows the model below (see also <https://www.embopress.org/page/journal/17444292/authorguide#dataavailability>). Please

note that the Data Availability Section is restricted to new primary data that are part of this study.

Data availability

-At EMBO Press we ask authors to provide source data for the main figures. Our source data coordinator will contact you to discuss which figure panels we would need source data for and will also provide you with helpful tips on how to upload and organize the files.

- Our journal encourages inclusion of *data citations in the reference list* to directly cite datasets that were re-used and obtained from public databases. Data citations in the article text are distinct from normal bibliographical citations and should directly link to the database records from which the data can be accessed. In the main text, data citations are formatted as follows: "Data ref: Smith et al, 2001". In the Reference list, data citations must be labeled with "[DATASET]". A data reference must provide the database name, accession number/identifiers and a resolvable link to the landing page from which the data can be accessed at the end of the reference. Further instructions are available at .

- We updated our journal's competing interests policy in January 2022 and request authors to consider both actual and perceived competing interests. Please review the policy <https://www.embopress.org/competing-interests> and update your competing interests if necessary.

Please use the heading "Disclosure statement and competing interests".

- All Materials and Methods need to be described in the main text using our 'Structured Methods' format. According to this format, the Methods section includes a Reagents and Tools Table (listing key reagents, experimental models, software and relevant equipment and including their sources and relevant identifiers) followed by a Methods and Protocols section describing the methods, ideally using a step-by-step protocol format. The aim is to facilitate adoption of the methodologies across labs.

Please download and fill our Reagents and Tools Table template (.docx), which you can find in our author guidelines:

<https://www.embopress.org/page/journal/17444292/authorguide#structuredmethods>.

-Regarding data quantification:

Please ensure to specify the name of the statistical test used to generate error bars and P values, the number (n) of independent experiments (please specify technical or biological replicates) underlying each data point and the test used to calculate p-values in each figure legend. Discussion of statistical methodology can be reported in the materials and methods section, but figure legends should contain a basic description of n, P and the test applied.

Graphs must include a description of the bars and the error bars (s.d., s.e.m.).

- Please provide a "standfirst text" summarizing the study in one or two sentences (approximately 250 characters, including space), three to four "bullet points" highlighting the main findings and a "synopsis image" (550px width and 400-600 px height, PNG format) to highlight the paper on our homepage.

Here are a couple of examples:

<https://www.embopress.org/doi/10.15252/msb.20199356>

<https://www.embopress.org/doi/10.15252/msb.20209475>

<https://www.embopress.org/doi/10.15252/msb.209495>

When you resubmit your manuscript, please download our CHECKLIST (<https://www.embopress.org/pb-assets/embo-site/EMBO%20Press%20Author%20Checklist-1642513524327.xlsx>) and include the completed form in your submission.

Please note that the Author Checklist will be published alongside the paper as part of the transparent process (<https://www.embopress.org/page/journal/17444292/authorguide#transparentprocess>).

If you feel you can satisfactorily deal with these points and those listed by the referees, you may wish to submit a revised version of your manuscript. Please attach a covering letter giving details of the way in which you have handled each of the points raised by the referees. A revised manuscript will be once again subject to review and you probably understand that we can give you no guarantee at this stage that the eventual outcome will be favorable.

I look forward to seeing a revised manuscript soon.

Yours sincerely,
Jingyi

Jingyi Hou, PhD
Senior Editor
Molecular Systems Biology

We realize that it is difficult to revise to a specific deadline. In the interest of protecting the conceptual advance provided by the work, we recommend a revision within 3 months (30th Jun 2025). Please discuss the revision progress ahead of this time with the editor if you require more time to complete the revisions. Use the link below to submit your revision:

IMPORTANT: When you send your revision, we will require the following items:

1. the manuscript text in LaTeX, RTF or MS Word format
2. a letter with a detailed description of the changes made in response to the referees. Please specify clearly the exact places in the text (pages and paragraphs) where each change has been made in response to each specific comment given
3. three to four 'bullet points' highlighting the main findings of your study
4. a short 'blurb' text summarizing in two sentences the study (max. 250 characters)
5. a 'thumbnail image' (550px width and max 400px height, Illustrator, PowerPoint or jpeg format), which can be used as 'visual title' for the synopsis section of your paper.
6. Please include an author contributions statement after the Acknowledgements section (see <https://www.embopress.org/page/journal/17444292/authorguide>)
7. Please complete the CHECKLIST available at (<https://bit.ly/EMBOPressAuthorChecklist>). Please note that the Author Checklist will be published alongside the paper as part of the transparent process (<https://www.embopress.org/page/journal/17444292/authorguide#transparentprocess>).
8. When assembling figures, please refer to our figure preparation guideline in order to ensure proper formatting and readability in print as well as on screen:
<https://bit.ly/EMBOPressFigurePreparationGuideline>
See also figure legend guidelines: <https://www.embopress.org/page/journal/17444292/authorguide#figureformat>
9. Please note that corresponding authors are required to supply an ORCID ID for their name upon submission of a revised manuscript (EMBO Press signed a joint statement to encourage ORCID adoption). (<https://www.embopress.org/page/journal/17444292/authorguide#editorialprocess>)
Currently, our records indicate that the ORCID for your account is 0000-0002-8364-1991.

Link Not Available

11. Include a Reagents and Tools Table as part of the Methods section, which can be downloaded from our author guidelines (<https://www.embopress.org/page/journal/17444292/authorguide#structuredmethods>)

*** PLEASE NOTE *** As part of the EMBO Press transparent editorial process initiative (see our Editorial at <https://dx.doi.org/10.1038/msb.2010.72>), Molecular Systems Biology publishes online a Review Process File with each accepted manuscripts. This file will be published in conjunction with your paper and will include the anonymous referee reports, your point-by-point response and all pertinent correspondence relating to the manuscript. If you do NOT want this File to be published, please inform the editorial office at msb@embo.org within 14 days upon receipt of the present letter.

Reviewer #1:

I have the revised version of the manuscript by Camacho and coworkers entitled "Convergence of aging- and rejuvenation-related epigenetic alterations on PRC2 targets". While the authors have convincingly addressed some of the issues raised in my initial review, I am still concerned about some of them. An important factor is still novelty, although I understand that novelty is less important for some journals than others. I am also not entirely satisfied by the depth of the analysis and changes introduced into this revised version, which overall seem insufficient. For some of the questions, the authors do not seem to specify which are the new analyses and plots, and these are hard to find in the manuscript. This should have been clear in the rebuttal and manuscript to help reviewers to properly evaluate the revised version.

Please find below some examples that require attention:

General comments:

1. The code deposited in Github only corresponds to the later ChIP-seq analysis. Code availability had been previously requested but has not been adequately addressed as of yet, complicating the revision of the manuscript.

2. The requested exploration of cell type deconvolution analyses was not performed, and no justification was given as to the lack of analyses.

3. As requested, the authors state that now they include scatter plots of correlation between genome-wide MML and NME in Supplementary Figure 2. But these plots are not shown in that figure.

4. Regarding statistical tests used for the boxplot figures (e.g. main Figure 1B, Supplementary Figure 1B). It seems surprising that some of the changes reach significance as the effect size is small and variability is high. Were the samples treated as independent biological replicates and the means or medians are compared via t-test, Wilcoxon...? Or were the samples pooled in a different type of statistical test? Only the first approach is valid when the authors have biological replicates in their data.

Specific comments:

5. With regard to figure 1:

d. Fig. 1b seems to be redundant with Fig. 1c-e. Also, they arbitrarily choose the most different or similar samples to plot

Author's response: 1c-e we wanted to emphasize that the OSKM treatment globally restores MML and NME to a state that resembles the young samples. We recognize they can seem redundant as global trends are already shown in Fig. 2 in the form of a histogram, including all samples.

Reviewer: I still see no value for Fig. 1c-e aside from the redundancy, the arbitrary choosing of specific samples is misleading.

f. Fig. 1c, all cases could be represented, as few are shown.

Author's response: Even though we only show a comparison between representative samples in the density plots, all the samples are represented in the histograms of Fig. 1b and the subsequent analyses and statistics for differential DNA methylation take into account all the available samples from the different groups

Reviewer: The authors should include all of these cases.

6. With regard to figure S1:

a. Correct me if I am wrong, but this figure does not show the same trend as Fig. 1c in islands, for MML. Please explain.

Author's response: Most CpG islands remain unmethylated in somatic cells (PMID 22641018). In Figure S1 we show CpG islands remain highly unmethylated during aging, with a very mild increase in MML in old compared to young samples. Indeed, this differs from the trend of global age-related hypomethylation we observe at the genome-wide level and reflects the majority of CpG islands may not show significant changes in MML during aging. As this is a summary plot describing a general tendency in such regions, it does not preclude that there are specific islands with significant changes in MML during aging. Although CpG islands in intragenic and gene body regions can have tissue-specific patterns of methylation and are prompt to change upon environmental exposures, CpG islands associated with transcription start sites rarely show tissuespecific methylation patterns and are less plastic (PMID 20613842). Instead, CpG shores, located as far as 2 kb from CpG islands, have highly conserved patterns of tissue-specific methylation (PMID 19151715) and we show MML in those are more prone to change during aging.

Reviewer: I don't think that "the majority of CpG islands may not show significant changes in MML during aging". It seems to me that they do, as there is a statistical test done, indicating significance "***", in Supplementary Figure 1B. The fact that CpG islands specifically increase MML with age (and that this seems also to be reverted with OSKM), as opposed to the rest of the genome, should be discussed.

b. Could you please distinguish promoters from gene body, exon, intron, intergenic regions?

Author's response: We define promoters as regions that are 2000 bp upstream and downstream from the transcriptional start site of the gene and have clarified that in the section on genomic features and annotations. Gene bodies do not include the first 2000 bp in the 5' of the gene that overlap with the defined promoter. In the same manner, intergenic regions do not include the 2000 bp upstream of the transcriptional start of a gene, not overlapping with defined promoters.

Reviewer: The authors should include in the Supplementary Figure 1B boxplots the additional category of promoters to see the values of MML and NME at those regions.

7. With regard to your statement: "We identified the most epigenetically discordant genes and evaluated for enrichments among the top 500 genes in each group-wise comparison using Gene Set Enrichment Analysis (GSEA)."

a. The authors identify top 500 genes by methylation. But, then, how is the GSEA done? Using a measure from the JSD? What's the background/universe for the enrichment analysis?

Author's response: As described in the Methods section, when performing comparisons between samples, we calculate JSD values for every non-overlapping 150 bp genomic bin in the genome. To have a representative JSD value for a promoter or gene body of each gene, we calculate the square root of the average of the squared JSD values within all analysis 150 bp bins that overlap with the promoter or gene body. As shown in Supplementary Tables S2, S3 and S7, we performed our analysis considering only promoters and also combining both promoters and gene bodies. In the Methods section we describe how we implement hypothesis testing to decide if a promoter or gene body is statistically different among 2 conditions based on JSD, generating a list of genes with adjusted p-values associated to the statistical differences that the corresponding promoter or gene body shows. We selected the top 500 genes that showed the most significant differences based on adjusted p-value. The GSEA was done using the software from the Broad Institute (PMID 16199517), which performs GSEA using a hypergeometric test and provides a p-adjusted value and an odds ratio. The background for the enrichment analysis was set to be all the genes in the mouse genome according to the GENCODE VM23 annotation, used by the TxDb.Mmusculus.UCSC.mm10.knownGene package in R, as that it is the number of genes considered in the DNA methylation differential analysis. We updated the Methods section to include a better description of the analyses performed.

Reviewer: I think that it may be a bit misleading to state that the authors performed "GSEA" as the method used was an hypergeometric test. It is true that they use the Broad Institute tool, but typically in the literature GSEA refers to the specific method which is described in the papers the authors cite (PMID 16199517). This should be more clarified in the manuscript. Also, please indicate versions of software used.

10. For Boxplots in general, what are the asterisks between the groups? To which test do they refer to?

Author's response: Differences in global distributions of MML and NME were statistically tested using the MannWhitney U test. Significant differences were indicated in histogram plots with an asterisk. We updated the Methods section including this information.

Reviewer: What is the meaning of the "***", please explicitly indicate in Methods (e.g. <0.001). It seems surprising that some of the changes reach significance as the effect size is small and variability is high. Were the samples treated as independent biological replicates? Or were the samples pooled in a different type of statistical test? Only the first approach is valid when the authors have biological replicates in their data. This is also relevant because there is a minimum sample size appropriate for Wilcoxon testing.

Reviewer #2:

The current revision has addressed most of the concerns of the reviewers. One issue that remains is the question of mechanism. Indeed, the authors have now conducted a new H3K27me3 ChIP-seq. However, the integration of this dataset with the previous DNA methylation data is far too superficial to draw any meaningful conclusions. In order to understand this better, the authors need to show:

- Basic statistics for the ChIP-seq: sequencing depth, alignments scores, number of mapped peaks, PCA and correlation plots to assess reproducibility
- Correlation of DNA methylation and H3K27me3 changes. One good option would be a meta-profile across TSS and at least a Venn diagram to show the fraction of differential peaks in H3K27me3 as well as in DNA methylation (the 98 identified genes need to be placed in context with the overall number of changes in both datasets). In such a Venn diagram, the transcriptome data could be integrated as well.
- the browser shots should show tracks of RNA-seq

Finally, the authors need to tone down their language with respect to causality. Yes, it appears that PRC2 plays a role in mediating the effects seen throughout rejuvenation. However, with two NGS datasets and no perturbation, it is impossible to draw any direct causation. This should be done throughout the manuscript.

Pre-decision cross-commenting

Reviewer #2

I do agree with reviewer 1 on code availability - this is a must and I would like to add this to my report as well. In addition, I also agree with reviewer 1 on the choice of Fig. 1c-e. The authors should show an average and or include all examples (could be as part of supplements and/or figshare)

We thank the reviewers for working with us on further improving our manuscript. The following are our responses and descriptions of corresponding changes to the paper:

Reviewer #1:

I have the revised version of the manuscript by Camacho and coworkers entitled "Convergence of aging- and rejuvenation-related epigenetic alterations on PRC2 targets". While the authors have convincingly addressed some of the issues raised in my initial review, I am still concerned about some of them. An important factor is still novelty, although I understand that novelty is less important for some journals than others. I am also not entirely satisfied by the depth of the analysis and changes introduced into this revised version, which overall seem insufficient. For some of the questions, the authors do not seem to specify which are the new analyses and plots, and these are hard to find in the manuscript. This should have been clear in the rebuttal and manuscript to help reviewers to properly evaluate the revised version.

Thanks for calling this to our attention. We have now provided both a clean and a tracked version of the manuscript which shows where we addressed criticisms and where we performed new analyses, and have also itemized the changes in the responses below.

General comments:

1. The code deposited in Github only corresponds to the later ChIP-seq analysis. Code availability had been previously requested but has not been adequately addressed as of yet, complicating the revision of the manuscript.

All code has been placed in Github: https://github.com/ocamach1/PRC2_aging_rejuvenation This includes scripts to:

- Analyze WGBS data:
 - Align and process WGBS data from raw fastq files.
 - Run informME pipeline: compute mean methylation level (MML), normalized entropy level (NME) for each WGBS sample, and Jensen-Shannon distance (JSD) between pairs of samples.
 - Rank epigenetically discordant genes when comparing samples under 2 conditions (i.e. old vs young) based on JSD.
 - Perform cell type deconvolution with MeDeCom.
 - Perform Student's t-test and Wilcoxon test to compare distributions of MML and NME across groups.
- Analyze ChIP-Seq data:
 - Align and process ChIP-Seq data from raw fastq files.
 - Perform peak calling with Sicer2 (for H3K27me3).
 - Perform peak calling with EDD (for H3K9me2).
 - Perform differential binding analysis with DiffBind.
- Analyze RNA-Seq data:
 - Align and process RNA-Seq data from raw fastq files.
 - Perform differential expression analysis with DESeq2.
- Perform over-representation analysis (ORA)
- Plots

2. The requested exploration of cell type deconvolution analyses was not performed, and no justification was given as to the lack of analyses.

As suggested, to explore the possibility that DNA methylation differences may be driven by heterogeneous cell-type composition in our dorsal skin samples, we performed a cell-type

deconvolution analysis. This was limited by the lack of available reference methylomes for mouse skin cell types, requiring the use of a reference-free approach. We used MeDeCom, an extensively used and well-validated method (Lutsik et al 2017), which decomposes the DNA methylation data into latent DNA methylation components (LMC) and determines their proportions in each sample, with the goal of inferring cell-type proportions in the samples. The text has been modified to include this analysis and results are provided in Fig. EV1C. The results are notable for fairly homogeneous methylation within samples, without evidence for a mixture driven by multiple LMCs. Rather, with aging or rejuvenation treatment, there is a shift in the overall methylation pattern driving a shift toward a different LMC. Importantly, since LMCs can be driven by biologically meaningful differences in methylation as a result of aging or rejuvenation itself, a shift in the dominant LMC does not imply a cell-type composition shift. Rather, the lack of a mixture of LMCs in each samples would argue against substantial cell-type heterogeneity driving the overall methylation findings.

Specific changes are found in Results (1st paragraph below), Methods (2nd paragraph below) and in new Figure EV1c:

Because whole skin comprises multiple cell types and DNA methylation differences may be driven by shifts in cell-type proportions, we performed a cell-type deconvolution analysis. In absence of reference methylomes for mouse skin cell types, we used a reference-free approach implemented by the MeDeCom software (Lutsik et al, 2017), which decomposes the DNA methylation data into latent DNA methylation components (LMC) and their proportions in each sample (Fig. EV1C), with the goal of inferring cell-type proportions in the samples. The deconvolution results revealed that most samples were dominated by a single LMC, indicating relatively homogeneous DNA methylation patterns within samples and arguing against substantial cell-type admixture. Rather, aging or rejuvenation treatment was associated with a shift in the overall DNA methylation pattern, leading to a transition toward a different LMC. Importantly, since LMCs can be driven by biologically meaningful differences in DNA methylation as a result of aging or rejuvenation itself, a shift in the dominant LMC does not imply a cell-type composition shift.

Cell type deconvolution from WGBS data. BED methyl files from whole dorsal skin WGBS samples were generated from deduplicated BAM files using the function 'methylation extractor' from Bismark v.0.22.2 (Krueger & Andrews, 2011) and used as input for downstream analyses. DNA methylation data was processed and filtered using RnBeads v.3.21 (Müller et al, 2019). All sites having read coverage lower than 5 in any of the samples, high and low coverage outliers, sites with missing values, annotated SNPs, and sites on the sex chromosomes were removed. In absence of reference DNA methylation data for cell types in mouse skin, we employed MeDeCom v.1.0.2 (Lutsik et al., 2017), a reference-free computational framework that uses regularized non-negative matrix factorization to decompose complex DNA methylation data into latent methylation components (LMC) and their proportions in each sample. We selected the 5000 most variably methylated CpGs across the samples for this analysis. Since we did not have prior knowledge on the expected number of underlying cell types to select, we resorted to the cross-validation procedure of MeDeCom. We chose 7 LMCs as the value of K at which the cross-validation error started to level out and selected $\lambda = 0.001$ (regularization parameter) as the point where the cross-validation error is still low, while the objective

value and RMSE substantially change, following the guidelines described by the authors of the MeDeCom software (Lutsik et al., 2017).

3. *As requested, the authors state that now they include scatter plots of correlation between genome-wide MML and NME in Supplementary Figure 2. But these plots are not shown in that figure.*

To clarify, we were referring to Supplementary Figure 2 of Jenkinson et al., *Nature Genetics* 2017 May;49(5):719-729, PMID 28346445. The figure, titled “Mean methylation level is not predictive of normalized methylation entropy,” presents genome-wide scatter plots of MML and NME for cancer-normal pairs analyzed in that paper, as well as scatter plots of dMML and dNME values. In general, although some NME values can be predicted from MML values, in general the NME cannot be inferred from the MML – notably, genomic units with the same MML values may be characterized by different NME values. Additionally, as shown in that figure as well as in Koldobskiy et al *Nat Biomed Eng* 2021 PMID 33859388, genomic units with no mean methylation difference (dMML values of zero) are not necessarily characterized by zero dNME values.

4. *Regarding statistical tests used for the boxplot figures (e.g. main Figure 1B, Supplementary Figure 1B). It seems surprising that some of the changes reach significance as the effect size is small and variability is high. Were the samples treated as independent biological replicates and the means or medians are compared via t-test, Wilcoxon...? Or were the samples pooled in a different type of statistical test? Only the first approach is valid when the authors have biological replicates in their data.*

The prior analysis compared methylation results at each genomic unit across groups, driving the very small p-values. In accordance with the reviewer’s suggestion, we have switched to using a simple two-sided independent Student’s t-test, carrying out comparisons using the average MML or NME value per sample and comparing groups treating the samples as independent biological replicates, as suggested (Young group: n = 3; Old untreated group: n = 4; Old treated group: n = 5). In the majority of cases, differences were still significant, but with more modest p-values. Significance is indicated in the figures. The new text in the Methods is reproduced here:

Differences in MML and NME between experimental conditions, assessed either genome-wide or across selected genomic regions, were evaluated using two complementary statistical analyses. First, we performed independent Student’s t-tests using one value per sample, obtained by averaging the MML or NME values across all 150-bp analysis regions genome-wide (or across tiles within the selected genomic regions). These per-sample averages were then compared between conditions, treating each mouse as an independent biological replicate (Young: n = 3; Old untreated: n = 4; Old treated: n = 5). Comparisons between young and old untreated mice were evaluated with two-sided tests. For comparisons between old untreated and old treated mice, we report both two-sided and one-sided p-values, with the one-sided tests reflecting our a priori expectation that OSKM treatment would reverse age-associated changes. Second, to evaluate the consistency and directionality of methylation changes across the genome, we performed a paired Wilcoxon signed-rank test on 150-bp analysis regions. For each region, the average MML or NME within each condition was calculated by aggregating across biological replicates, and the resulting per-region values were compared in a paired design. We report the Wilcoxon pseudo-median (e), which provides a robust estimate of the central tendency of the paired differences in MML or NME across 150-bp analysis regions.

This is also represented in Figure 1B, with the following legend:

Boxplots of genome-wide distributions of MML and NME. The median (middle quartile) marks the mid-point of the data and is shown by the line that divides the box into two parts. The box indicates the interquartile range (IQR). The whiskers indicate 1.5 x IQR. To compare groups, complementary tests were performed: two-tailed and one-tailed Student's t-test (2T, 1T) on the average signal per sample, treating each biological replicate as one data point (Young, n = 3; Old, n = 4; Old+OSKM, n = 5); and a Wilcoxon signed-rank test (W) on tiled 150 bp genomic regions, where the signal was averaged across replicates within each group and compared tile-by-tile across the genome. All Wilcoxon signed-rank tests yielded $p < 2.2 \times 10^{-16}$ and pseudo-medians (e) are reported. MML comparisons: Old vs Young (2T: $p = 0.041$; W: -0.018); Old+OSKM vs Old (2T: $p = 0.053$, 1T: $p = 0.027$; W: $e = -0.017$). NME comparisons: Old vs Young (2T: $p = 0.042$; W: $e = 0.026$); Old+OSKM vs Old (2T: $p = 0.031$, 1T: $p = 0.016$; W: $e = 0.028$).

Specific comments:

5. *With regard to figure 1:*

d.Fig. 1b seems to be redundant with Fig. 1c-e. Also, they arbitrarily choose the most different or similar samples to plot. Author's response: 1c-e we wanted to emphasize that the OSKM treatment globally restores MML and NME to a state that resembles the young samples. We recognize they can seem redundant as global trends are already shown in Fig. 2 in the form of a histogram, including all samples. Reviewer: I still see no value for Fig. 1c-e aside from the redundancy, the arbitrary choosing of specific samples is misleading.

We felt that the density plots help convey the distribution of data and the shift of MML and NME between groups. To avoid only showing some arbitrarily selected comparisons, we now provide all density plots in the Appendix, Figure 1.

f.Fig, 1c, all cases could be represented, as few are shown. Author's response: Even though we only show a comparison between representative samples in the density plots, all the samples are represented in the histograms of Fig. 1b and the subsequent analyses and statistics for differential DNA methylation take into account all the available samples from the different groups

Reviewer: The authors should include all of these cases.

This has been done and is now included in the Appendix, Figure 1.

6. *With regard to figure S1:*

a. Correct me if I am wrong, but this figure does not show the same trend as Fig. 1c in islands, for MML. Please explain. Author's response: The reviewer raises an interesting point and we have made an effort to emphasize this more in the manuscript. Most CpG islands remain unmethylated in somatic cells (PMID 22641018). In Figure S1 we show CpG islands remain highly unmethylated during aging, with a very mild increase in MML in old compared to young samples. Indeed, this differs from the trend of global age-related hypomethylation we observe at the genome-wide level, driven largely by non-island genomic regions, and reflects that the majority of CpG islands may not show significant changes in MML during aging. As this is a summary plot describing a general tendency in such regions, it does not preclude that there are specific islands with significant changes in MML during aging. Although CpG islands in intragenic and gene body regions can have tissue-specific patterns of methylation and change upon

environmental exposures, CpG islands associated with transcription start sites rarely show tissue-specific methylation patterns and are less plastic (PMID 20613842). Instead, CpG shores, located as far as 2 kb from CpG islands, have highly conserved patterns of tissue-specific methylation (PMID 19151715) and we show MML in those are more prone to change during aging.

*Reviewer: I don't think that "the majority of CpG islands may not show significant changes in MML during aging". It seems to me that they do, as there is a statistical test done, indicating significance "**", in Supplementary Figure 1B. The fact that CpG islands specifically increase MML with age (and that this seems also to be reverted with OSKM), as opposed to the rest of the genome, should be discussed.*

Using t-tests as the reviewer requested, the differences in MML over all CpG islands with aging (Fig EV1B) were not significant. However, we thank the reviewer for noting this potentially important trend and include it in the revised text. We further explored methylation changes at CpG islands that are marked by H3K27me3 in integrating our ChIP-seq and methylation data. Interestingly, among sites that lose H3K27me3, CpG islands within promoters exhibit a slight gain in MML but a more pronounced gain of NME (Figure EV3D). In the discussion, we speculate about the differing methylation profiles of CpG islands versus heterochromatin regions with aging, which parallels findings in cancer epigenomes (large hypomethylated blocks and focal hypermethylated CpG islands).

We discuss these findings in the text as reproduced here (1st two paragraphs in Results, third in discussion):

A similar trend was consistent when examining MML and NME distributions over selected genomic features, with the exception of CpG islands which exhibit a trend toward increased MML with aging (Fig. EV1B).

A further comparison was performed for Young vs Old and Old vs rejuvenated whole skin by stratifying promoters containing CpG islands according to their H3K27me3 dynamics (unmarked, retained, or marked with loss between the compared groups), and quantifying associated differences in CpG island-localized MML and NME. Among the promoters that exhibited an age-related loss of H3K27me3, there was a marked increase in NME and a slight increase in MML in both aging and rejuvenation comparisons (Fig. EV3D).

This profound change of DNA methylation stochasticity at PRC2 sites and large-scale heterochromatin blocks in aging has parallels to cancer epigenomes, which are marked by large hypomethylated blocks and focal hypermethylated CpG islands (Feinberg et al, 2016).

b. Could you please distinguish promoters from gene body, exon, intron, intergenic regions? Author's response: We define promoters as regions that are 2000 bp upstream and downstream from the transcriptional start site of the gene and have clarified that in the section on genomic features and annotations. Gene bodies do not include the first 2000 bp in the 5' of the gene that overlap with the defined promoter. In the same manner, intergenic regions do not include the 2000 bp upstream of the transcriptional start of a gene, not overlapping with defined promoters.

We have clarified these definitions in the Methods, as requested:

Files and tracks bear genomic coordinates for mm10. We used the R package 'TxDb.Mmusculus.UCSC.mm10.knownGene' to define genes, exons, and introns. We obtained CpG island annotations from the UCSC Genome Browser. We defined the

promoter region of a gene as the 4-kb window centered at its transcription start site and determined the gene body region to be the remainder of the gene.

Reviewer: The authors should include in the Supplementary Figure 1B boxplots the additional category of promoters to see the values of MML and NME at those regions.

This has been done in Supplementary Figure EV1.

7. With regard to your statement: "We identified the most epigenetically discordant genes and evaluated for enrichments among the top 500 genes in each group-wise comparison using Gene Set Enrichment Analysis (GSEA)."

a. The authors identify top 500 genes by methylation. But, then, how is the GSEA done? Using a measure from the JSD? What's the background/universe for the enrichment analysis? Author's response: As described in the Methods section, when performing comparisons between samples, we calculate JSD values for every non-overlapping 150 bp genomic bin in the genome. To have a representative JSD value for a promoter or gene body of each gene, we calculate the square root of the average of the squared JSD values within all analysis 150 bp bins that overlap with the promoter or gene body. As shown in Supplementary Tables S2, S3 and S7, we performed our analysis considering only promoters and also combining both promoters and gene bodies. In the Methods section we describe how we implement hypothesis testing to decide if a promoter or gene body is statistically different among 2 conditions based on JSD, generating a list of genes with adjusted p-values associated to the statistical differences that the corresponding promoter or gene body shows. A detailed description of this method and the adjusted p-values is cited: Jenkinson et al BMC Bioinformatics 2019 (PMID: 30961526) We selected the top 500 genes that showed the most significant differences based on adjusted p-value. The over-representation analysis was performed using Fisher's exact test, providing a p-adjusted value and an odds ratio. The background for over-representation analysis for DNA methylation was set to be all the genes in the mouse genome according to the GENCODE VM23 annotation, used by the TxDb.Mmusculus.UCSC.mm10.knownGene package in R, as that it is the number of genes considered in the DNA methylation differential analysis. We updated the Methods section to include a better description of the analyses performed.

Reviewer: I think that it may be a bit misleading to state that the authors performed "GSEA" as the method used was an hypergeometric test. It is true that they use the Broad Institute tool, but typically in the literature GSEA refers to the specific method which is described in the papers the authors cite (PMID 16199517). This should be more clarified in the manuscript. Also, please indicate versions of software used.

We have modified the text to clarify the methods for over-representation analysis. This is reproduced here:

Over-representation analysis

Over-representation analysis (ORA) was performed using the `fora` function from the `fgsea` R package (v3.22), which applies Fisher's exact tests to evaluate whether a given gene list is over-represented within curated gene sets (C2 collection) from the MSigDB database (Liberzon et al., 2015). For the ORA of differentially methylated genes, we used the top 500 genes ranked by JSD magnitude (and significant JSD-based epigenetic

discordance; adjusted $p < 0.05$) and tested their over-representation across all the gene sets from the C2 collection. The background (universe) was defined as all genes considered in the ranking by JSD magnitude (Datasets EV1 and EV2), corresponding to the complete set of genes annotated in the TxDb.Mmusculus.UCSC.mm10.knownGene package. For the ORA of differentially expressed genes, genes with adjusted $p < 0.05$ and shrunken \log_2 fold change < -1.5 or > 1.5 in the DESeq2 analysis were tested for over-representation among the top five gene sets from the C2 collection that ranked highest in the JSD-based ORA. The background was defined as all genes captured by RNA-seq and retained in the differential expression analysis with DESeq2 (Datasets EV3 and EV4).

10. For Boxplots in general, what are the asterisks between the groups? To which test do they refer to? Author's response: Differences in global distributions of MML and NME were statistically tested using the MannWhitney U test. Significant differences were indicated in histogram plots with an asterisk. We updated the Methods section including this information.

Reviewer: What is the meaning of the “”, please explicitly indicate in Methods (e.g. <0.001). It seems surprising that some of the changes reach significance as the effect size is small and variability is high. Were the samples treated as independent biological replicates? Or were the samples pooled in a different type of statistical test? Only the first approach is valid when the authors have biological replicates in their data. This is also relevant because there is a minimum sample size appropriate for Wilcoxon testing.*

This has been modified per the reviewer's suggestion as addressed in point 4 above. P-values are provided in the figure legends rather than using asterisks/symbols to avoid confusion, and the statistical tests used are explicitly indicated.

Reviewer #2:

The current revision has addressed most of the concerns of the reviewers. One issue that remains is the question of mechanism. Indeed, the authors have now conducted a new H3K27me3 ChIP-seq. However, the integration of this dataset with the previous DNA methylation data is far too superficial to draw any meaningful conclusions. In order to understand this better, the authors need to show:

- Basic statistics for the ChIP-seq: sequencing depth, alignments scores, number of mapped peaks, PCA and correlation plots to assess reproducibility

Basic statistics for WGBS and ChIP-seq, PCA, and correlation plots have now been included in Table EV1 and Figure EV3A,B and EV5D,E.

- Correlation of DNA methylation and H3K27me3 changes. One good option would be a meta-profile across TSS and at least a Venn diagram to show the fraction of differential peaks in H3K27me3 as well as in DNA methylation (the 98 identified genes need to be placed in context with the overall number of changes in both datasets). In such a Venn diagram, the transcriptome data could be integrated as well.

As recommended, we have expanded the analysis of convergence between H3K27me3 and DNA methylation alterations.

As suggested, to evaluate the relationship between DNA methylation alterations and H3K27me3 occupancy, we generated metaplots over promoter and gene body regions comparing H3K27me3 peaks and JSD between young and old from both dorsal skin and purified epidermis (Fig. EV4C-G) comparisons. Importantly, maximal H3K27me3 colocalized with high JSD regions at gene promoters, where the pattern of aging-related H3K27me3 was most pronounced (Fig. 3E). As suggested, we also provide a Venn diagram (Fig. 3C) to show the overlap of differential H3K27me3 and DNA methylation between groups, providing added context to the identified core gene set. These results are now described in the Results section “Convergent H3K27me3 and DNA Methylation Alterations Define a Core PRC2-Regulated Gene Set in Aging and Rejuvenation,” as reproduced below:

Importantly, there was a significant overlap between genes displaying differential H3K27me3 peaks by native ChIP between young and old in isolated epidermis and differentially DNA methylated genes (significant JSD magnitude over the promoter and gene body) with aging (OR = 2.14, $p = 8.05 \times 10^{-19}$) and partial reprogramming (OR = 2.19, $p = 6.41 \times 10^{-18}$) in dorsal skin (epidermis and dermis combined). The association was stronger when the overlap was restricted to genes with promoter-localized changes in H3K27me3 and DNA methylation (aging: OR = 2.86, $p = 7.50 \times 10^{-16}$; partial reprogramming: OR = 3.25, $p = 7.00 \times 10^{-17}$). A further comparison was performed for Young vs Old and Old vs rejuvenated whole skin by stratifying promoters containing CpG islands according to their H3K27me3 dynamics (unmarked, retained, or marked with loss between the compared groups), and quantifying associated differences in CpG island-localized MML and NME. Among the promoters that exhibited an age-related loss of H3K27me3, there was a marked increase in NME and a slight increase in MML in both aging and rejuvenation comparisons (Fig. EV3D).

We aimed to identify a core set of genes identified by our multimodal analysis as those with significant DNA methylation discordance (defined as significant JSD magnitude over promoter and gene body) during aging, significant DNA methylation discordance with partial reprogramming, and significant change in H3K27me3 distribution during aging. This produced an overlap set of 100 genes (Fig. 3C, Dataset EV6), representing a core set of PRC2 regulated genes that are epigenetically altered during aging and rejuvenation. Prominent examples within this core set include Sox11, Lin28b and Galnt13 (Fig. 3D and Fig. EV3E).

These changes are also reflected in Figure 3E and Figure EV4 as copied below:

Figure 3E. E. Metagene profiles of H3K27me3 occupancy from purified epidermis (top) and DNA methylation discordance measured as JSD from whole skin (bottom) during aging. Profiles were computed across the set of genes exhibiting significant aging-associated DNA methylation discordance (ranked by JSD magnitude over promoters and gene bodies) in whole skin. Gene bodies were scaled to a uniform length and flanked by ± 2 kb of unscaled upstream and downstream sequence. Thin lines represent individual replicates for H3K27me3 ChIP, while for JSD they represent 4 comparisons (each old vs a representative young (Y-3) sample)). Bold lines represent the average signal. This profile indicates coordinated promoter-proximal loss of H3K27me3 and elevated DNA methylation discordance with age.

Figure EV4. C-G. Metaplots of H3K27me3 occupancy and DNA methylation discordance (JSD) during aging and rejuvenation. Gene-level plots (D, F) show scaled gene bodies

flanked by ± 2 kb, while TSS-centered plots (C, E, G) show promoter-proximal regions spanning ± 4 kb around transcription start sites. Thin lines represent individual replicates for H3K27me3 ChIP, while for JSD they represent individual pairwise comparisons, where each old sample is compared against a representative control sample (aging in whole skin: Old vs Y-3; rejuvenation in whole skin: Old+OSKM vs O-2; aging in epidermis: Old vs Y-1). Bold lines represent the average signal. C. TSS-centered metaplot for the set of genes with significant promoter-restricted JSD-based methylation discordance during aging in whole skin. D. Gene-level metaplot for the set of genes with significant JSD-based methylation discordance over promoters and gene bodies in partial reprogramming in whole skin. E. TSS-centered metaplot for the set of genes with significant promoter-restricted JSD-based methylation discordance in partial reprogramming in whole skin. F. Gene-level metaplot for the set of the top 1000 JSD-ranked genes (based on discordance over promoter and gene body) during aging in purified epidermis. G. TSS-centered metaplot for the set of the top 1000 JSD-ranked genes (based on discordance over promoter) during aging in purified epidermis.

Additionally, from H3K9me2 native ChIP analysis, we also noted a striking overlap of differentially methylated regions and H3K9me2 LOCK regions. This is described in a new text section reproduced here:

We previously identified large organized K9-modification (LOCK) domains marked by H3K9me2 (Wen *et al.*, 2009). While age and sun exposure-related DNA hypomethylated blocks have been shown to overlap LOCKs, to our knowledge there is no known association between differential DNA methylation and LOCKs in aging itself, and DNA methylation and H3K9me2 have not been measured directly in the same tissues in aging (Vandiver *et al.*, 2015). To address this question, we used a contemporary methylation-block finding method ((Korthauer *et al.*, 2019), see Methods) on purified old and young epidermis, identifying 4,707 block differentially methylated regions (DMRs) (q -value < 0.05), 4,705 of which were hypomethylated in old compared to young mice (Dataset EV8). We also performed native ChIP-Seq for H3K9Me2 and identified LOCKs with an average width of 1.3 Mb (Dataset EV9, Fig. EV5D,E). The LOCKs were largely unchanged during aging, with few (9/795; FDR < 0.05) differential regions. Remarkably, we found block DMRs and H3K9me2-marked LOCKs co-localized significantly (p -value = 9.9×10^{-5} , Z -score = 62.39): 4106/4707 (87.23%) of block DMRs overlapped a LOCK and 551/795 (69.31%) of LOCKs overlapped at least one block DMR (Fig. 3F; also showing high degree of overlap with dNME), suggesting that LOCKs define regions where DNA methylation is lost in aging. There was no significant overlap between H3K27me3 peaks and H3K9me2 LOCKs ($p = 1$, $Z = -97.77$) or block DMRs ($p = 1$, $Z = -64.22$).

Finally, the authors need to tone down their language with respect to causality. Yes, it appears that PRC2 plays a role in mediating the effects seen throughout rejuvenation. However, with two NGS datasets and no perturbation, it is impossible to draw any direct causation. This should be done throughout the manuscript.

We appreciate this comment and have removed language suggesting causality of these findings in the manuscript. For example, the Discussion is more explicit on the need for further mechanistic and functional studies, e.g.:

In all, our analysis of H3K27me3 alterations and DNA methylation entropy changes during aging and rejuvenation may suggest a mechanistic influence of PRC2 that warrants further investigation.

Pre-decision cross-commenting

Reviewer #2

I do agree with reviewer 1 on code availability - this is a must and I would like to add this to my report as well. In addition, I also agree with reviewer 1 on the choice of Fig. 1c-e. The authors should show an average and or include all examples (could be as part of supplements and/or figshare)

We have deposited all code in github, and have provided results for all individual samples in the Appendix.

16th Jan 2026

Manuscript Number: MSB-2025-12937R

Title: Convergence of aging- and rejuvenation-related epigenetic alterations on PRC2 targets

Author: Oscar Camacho

Michael Koldobskiy

Pradeep Reddy

Atharv Oak

Yuxiang Sun

Kenna Sherman

Juan Carlos Izpisua Belmonte

Andrew Feinberg

Dear Dr Feinberg,

Thank you for sending us your revised manuscript. We have now heard back from the two reviewers who were asked to re-evaluate your study. As you will see, the reviewers are satisfied with the modifications made. Before we can formally accept your manuscript, we would ask you to address the following editorial-level issues:

1. Remove the "Author's contribution" section from the manuscript file.
2. In the Data Availability section, remove the reviewer token and ensure that all datasets will be made publicly available upon acceptance of the manuscript.
3. Expanded View (EV) tables and datasets: Please note that none of the files should be zipped, and each legend should be included as a separate sheet/tab within the same Excel file. Table EV1 is rather complex and should therefore be uploaded as a Dataset EV# file. The correct file type is Dataset (not Expanded View Content). Please update the nomenclature and callouts accordingly.
4. Appendix: Please provide figure legends for each figure and place each legend directly below its corresponding figure.
5. Please address the following issues related to figure legends identified by our data editor:
 - Please note that the box plots need to be defined in terms of minima, maxima, centre, bounds of box and whiskers, and percentile in the legend of figure EV2 A
 - Please note that information related to n is missing in the legends of figures 3A, EV2 A, EV4 B
 - Please note that for heatmap present in figures EV3 B, EV5 E, a numbered scale bar is not provided. This needs to be rectified.

Thank you for submitting this interesting paper to Molecular Systems Biology.

Kind regards,

Jingyi

Jingyi Hou, PhD

Senior Editor

Molecular Systems Biology

If you do choose to resubmit, please click on the link below to submit the revision online before 15th Feb 2026.

*** PLEASE NOTE *** As part of the EMBO Press transparent editorial process initiative (see our Editorial at <https://dx.doi.org/10.1038/msb.2010.72> , Molecular Systems Biology will publish online a Review Process File to accompany accepted manuscripts. When preparing your letter of response, please be aware that in the event of acceptance, your cover letter/point-by-point document will be included as part of this File, which will be available to the scientific community. More information about this initiative is available in our Instructions to Authors. If you have any questions about this initiative, please contact the editorial office (msb@embo.org).

Reviewer #1:

The authors have substantially improved the manuscript by adequately addressing the reviewers' criticisms. We have no further comments.

Reviewer #2:

The authors have addressed my concerns in this round of revision and I do not have any further comments.

All minor editorial requests have been addressed by the authors.

21st Jan 2026

Manuscript number: MSB-2025-12937RR

Title: Convergence of aging- and rejuvenation-related epigenetic alterations on PRC2 targets

Dear Andy,

Thank you again for sending us your revised manuscript. We are now satisfied with the modifications made and I am pleased to inform you that your paper has been accepted for publication.

You may qualify for financial assistance for your publication charges - either via a Springer Nature fully open access agreement or an EMBO initiative. Check your eligibility: <https://link.springer.com/journal/44320/how-to-publish-with-us>

Kind regards,
Jingyi

Jingyi Hou, PhD
Senior Editor
Molecular Systems Biology

>>> Please note that it is Molecular Systems Biology policy for the transcript of the editorial process (containing referee reports and your response letter) to be published as an online supplement to each paper. If you do NOT want this, you will need to inform the Editorial Office via email immediately. More information is available here: <https://link.springer.com/partners/embo-press/editorial-policies#Peer%20review>